# PPIA dictates NRF2 stability to promote lung cancer progression

Weiqiang Lu [1,2,12] ✉, Jiayan Cui[1,12], Wanyan Wang[1,12], Qian Hu[1,12], Yun Xue [3,4,12], Xi Liu[1], Ting Gong[1], Yiping Lu[1], Hui Ma[1], Xinyu Yang[2], Bo Feng[5], Qi Wang [6,7], Naixia Zhang[8], Yechun Xu [8], Mingyao Liu [2], Ruth Nussinov [9,10], Feixiong Cheng [11], Hongbin Ji [3,4] & Jin Huang [1] ✉

Nuclear factor erythroid 2-related factor 2 (NRF2) hyperactivation has been established as an oncogenic driver in a variety of human cancers, including non-small cell lung cancer (NSCLC). However, despite massive efforts, no specific therapy is currently available to target NRF2 hyperactivation. Here, we identify peptidylprolyl isomerase A (PPIA) is required for NRF2 protein stability. Ablation of PPIA promotes NRF2 protein degradation and blocks NRF2-driven growth in NSCLC cells. Mechanistically, PPIA physically binds to NRF2 and blocks the access of ubiquitin/Kelch Like ECH Associated Protein 1 (KEAP1) to NRF2, thus preventing ubiquitin-mediated degradation. Our X-ray co-crystal structure reveals that PPIA directly interacts with a NRF2 interdomain linker via a *trans*-proline 174-harboring hydrophobic sequence. We further demonstrate that an FDA-approved drug, cyclosporin A (CsA), impairs the interaction of NRF2 with PPIA, inducing NRF2 ubiquitination and degradation. Interestingly, CsA interrupts glutamine metabolism mediated by the NRF2/KLF5/SLC1A5 pathway, consequently suppressing the growth of NRF2-hyperactivated NSCLC cells. CsA and a glutaminase inhibitor combination therapy significantly retard tumor progression in NSCLC patient-derived xenograft (PDX) models with NRF2 hyperactivation. Our study demonstrates that targeting NRF2 protein stability is an actionable therapeutic approach to treat NRF2-hyperactivated NSCLC.

The transcription factor NRF2 (nuclear factor erythroid 2-related factor 2) and its negative regulator KEAP1 (Kelch Like ECH Associated Protein 1) contribute profoundly to the maintenance of the cellular redox homeostasis[1]. Under oxidative and electrophilic stress, NRF2 is released from KEAP1 and translocated to the nucleus to initiate the transcription of the antioxidant program[2]. This transient and inducible response was recognized originally as a protective mechanism against tumorigenesis[3]. However, many studies have subsequently indicated that constitutive NRF2 activation exhibits potent pro-tumorigenic properties in many types of cancer[4], including lung cancer.

Pan-cancer genome analysis of The Cancer Genome Atlas (TCGA) revealed that *NRF2* activation was particularly enhanced in non-small cell lung cancer (NSCLC)[5]. Genomic alterations that activated the NRF2 pathway ranked as the third most common in lung adenocarcinoma (LUAD, 25%), largely due to loss-of-function mutations in *KEAP1*[6]. Also, *KRAS* oncogenic mutations, the most prevalent oncogenic drivers in LUAD, potentiate transcriptional activation of *NRF2*[7]. *KEAP1* mutations co-exist in approximately 20% of *KRAS*-mutant LUAD[8]. Activated NRF2 not only promotes lung tumorigenesis through maintaining redox homeostasis and detoxification of reactive oxygen species (ROS)[7], but also facilitates aggressive lung cancer via multiple pro-metastasis factors[6,9]. Collectively, these comprehensive studies have established abnormal NRF2 activation as a pivotal oncogenic driver in lung cancer, implying a potential benefit of targeting NRF2 in cancer therapy.

---

However, targeted therapies that directly inhibit NRF2 remain inaccessible, emphasizing the necessity of exploring an alternative strategy.

In this study, we explore the innovative idea that NRF2 protein stability per se may (finally) enable a therapeutic approach for NRF2-hyperactivated NSCLC. Our biochemical, structural, and therapeutic model findings collectively reveal a proline-dependent regulatory mechanism for NRF2 stability, and we demonstrate that this mechanism can be pharmacologically targeted in NSCLC with NRF2 hyperactivation.

## Results

### Identification and validation of Cyclosporin A (CsA) as a chemical inducer of targeted NRF2 protein degradation

Given the known recalcitrance of directly inhibiting transcription factor NRF2[10], we adopted an alternative targeting strategy – exploiting the protein degradation machineries of NRF2. We first investigated NRF2 protein expression in 17 NSCLC cell lines. We detected a significant increase of NRF2 protein levels in nuclear extracts of *KEAP1* and/or *KRAS* mutant cell lines compared to corresponding wildtype (WT) cell lines (Table 1, Fig. 1A, B), which was concordant with previously studies[11,12]. To explore potential chemical inducer of NRF2 protein degradation, we generated a fluorescence-based sensor DsRed-IRES-EGFP-NRF2 (RIG-NRF2) for monitoring NRF2 stability in NSCLC cells (Fig. 1C). Briefly, we fused NRF2 to the fluorescent indicator protein EGFP and used DsRed as an internal control in a bicistronic lentiviral vector pLenti-DsRed-IRES-EGFP[13,14]. So, the EGFP/DsRed ratio can reflect the relative stability of NRF2. The resultant RIG-NRF2 was then stably transfected into NSCLC cell line A549 (Fig. 1C), and a monoclonal cell population was generated by limiting dilution.

An in-house library of 805 compounds (including clinical or approved drugs) was screened to identify potential compounds that can regulate NRF2 protein stability using the RIG-NRF2-expressing cell line above. The percentage of the decreased level of NRF2 was calculated by comparing that of the vehicle control (0%) with that of the cycloheximide (CHX)-treated group (100%) (Fig. 1D). Two known NRF2 activators Ki696 and ML334 were used as controls[15,16]. This screen revealed that the macrocyclic peptide drug cyclosporin A (CsA) exerted the strongest NRF2 degradation promoting effect among the library drugs (Fig. 1D and Supplementary Fig. 1A). We next examined CsA's effects on endogenous NRF2 in two representative NRF2-hyperactivated NSCLC cell lines (A549 and H2030). Immunoblotting analysis showed that CsA treatment reduced the NRF2 protein level in both cell lines (Supplementary Fig. 1B, C). In contrast, Q-PCR analysis revealed that mRNA level of NRF2 was not affected by CsA treatment (Supplementary Fig. 1D). The reduction of NRF2 protein level in CsA-treated cells was reversed by proteasome inhibition with MG132 (Supplementary Fig. 1E). CsA-treated cells exhibited a profound higher level of ubiquitinated NRF2 upon MG132 treatment, as compared with vehicle-treated cells (Supplementary Fig. 1F). These findings indicate that the ubiquitin-proteasome pathway is required for the degradation of NRF2 induced by CsA treatment. Intriguingly, we also observed a decrease of NRF2[T80K], a gain-of-function mutant of NRF2, upon CsA treatment (Supplementary Fig. 1G).

We further found that CsA treatment significantly repressed NRF2-driven luciferase activity and inhibited the expression of two classical NRF2 target genes, heme oxygenase 1 (*HMOX1*) and NAD(P)H:quinone oxidoreductase 1 (*NQO1*) in NRF2-hyperactivated A549 and H2030 cells, but not in H1650 and H1975 cells with low basal NRF2 level (Supplementary Fig. 1H-1J). Together, these results demonstrate that the FDA-approved drug CsA can induce the decrease of NRF2 protein levels in NRF2-hyperactivated NSCLC cells.

### CsA impedes NRF2-dependent growth of NSCLC cells

We next used CsA as a probe to delineate the functional contribution of NRF2 degradation in NSCLC. To this end, we monitored the potential anti-proliferative activities of CsA in NSCLC cell lines. Intriguingly, *KEAP1/KRAS* co-mutant cell lines, excepting for H1944, with augmented NRF2 protein levels were highly sensitive to CsA; *KEAP1* or *KRAS* single mutant cell lines partially responded to CsA; however, the corresponding WT cell lines and EBC-1 cells harboring heterogeneous D77V mutation with a low basal NRF2 level were largely resistant to CsA treatment (Fig. 1E, Supplementary Fig. 2A, B, Supplementary Table 1). Correlated with these results, CsA treatment profoundly reduced NRF2 protein levels in NRF2-hyperactivated NSCLC cells (Fig. 1F). We further found that siRNA-mediated knockdown of *NRF2* impaired the anti-proliferative capacity of CsA in NRF2-hyperactivated A549 and H2030 cells (Supplementary Fig. 2C). Overexpression of NRF2 or its gain-of function mutant (NRF2[T80K]) sensitized H1650 and H1975 cells, which have low basal NRF2 levels, to CsA treatment (Supplementary Fig. 2C). In line with these results, pharmacological activation of NRF2 by Ki696 and ML334 rendered sensitivity to CsA treatment in these two cell lines (Supplementary Fig. 2D). Together, these results demonstrate that CsA can suppress the NRF2-dependent growth and proliferation of NSCLC cells.

### Peptidylprolyl isomerase A (PPIA) is required for CsA-induced reduction of NRF2 levels and NSCLC cell proliferation

Peptidylprolyl isomerase A (PPIA, or CypA) is a specific cytosolic binding protein for CsA and is responsible for CsA's immunomodulating effects[17]. It has been shown to catalyze the *cis-trans* isomerization of proline residues and to regulate protein folding and stability[18]. These ideas led us to speculate that PPIA may be involved in the CsA-mediated promotion of NRF2 degradation. Pursuing this hypothesis, we knocked out (KO) the *PPIA* gene from A549 and H2030 using CRISPR/Cas9 genome editing. *PPIA*-knockout (*PPIA*-KO) resulted in profound reduction in the NRF2 protein levels, whereas showed no significant changes in mRNA levels of NRF2, in both KO cell lines (Fig. 2A and Supplementary Fig. 3A, B). However, CsA treatment caused no reduction in the NRF2 protein levels in the *PPIA*-KO cells (Fig. 2B and Supplementary Fig. 3C). We have demonstrated that CsA can induce NRF2 degradation via ubiquitin-proteasome system (Supplementary Fig. 1E, F). In line with this observation, *PPIA*-KO cells have a profound increased level of ubiquitinated NRF2 upon MG132 treatment compared with the unedited control cells (Fig. 2C and Supplementary Fig. 3D). These results demonstrate that PPIA is required for CsA-triggered degradation of the NRF2 protein.

**Table 1 | *KEAP1*, *NRF2* and *KRAS* gene mutations among various NSCLC cell lines**

| Gene Status | | | |
|---|---|---|---|
| Cell lines | KEAP1 | NRF2 | KRAS |
| A549 | p.G333C | NO. | p.G12S |
| EBC-1 | NO. | p.D77V | NO. |
| H1299 | NO. | NO. | NO. |
| H1395 | NO. | NO. | NO. |
| H1650 | NO. | NO. | NO. |
| H1703 | NO. | NO. | NO. |
| H1944 | p.R272L | NO. | p.G13D |
| H1975 | NO. | NO. | NO. |
| H2030 | p.V568F | NO. | p.G12C |
| H2122 | p.A170_R204del | NO. | p.G12C |
| H23 | p.Q193H | NO. | p.G12C |
| H292 | NO. | NO. | NO. |
| H358 | NO. | NO. | p.G12C |
| H441 | NO. | NO. | p.G12V |
| H460 | p.D236H | NO. | p.Q61H |
| H522 | NO. | NO. | NO. |
| H838 | p.E444* | NO. | NO. |

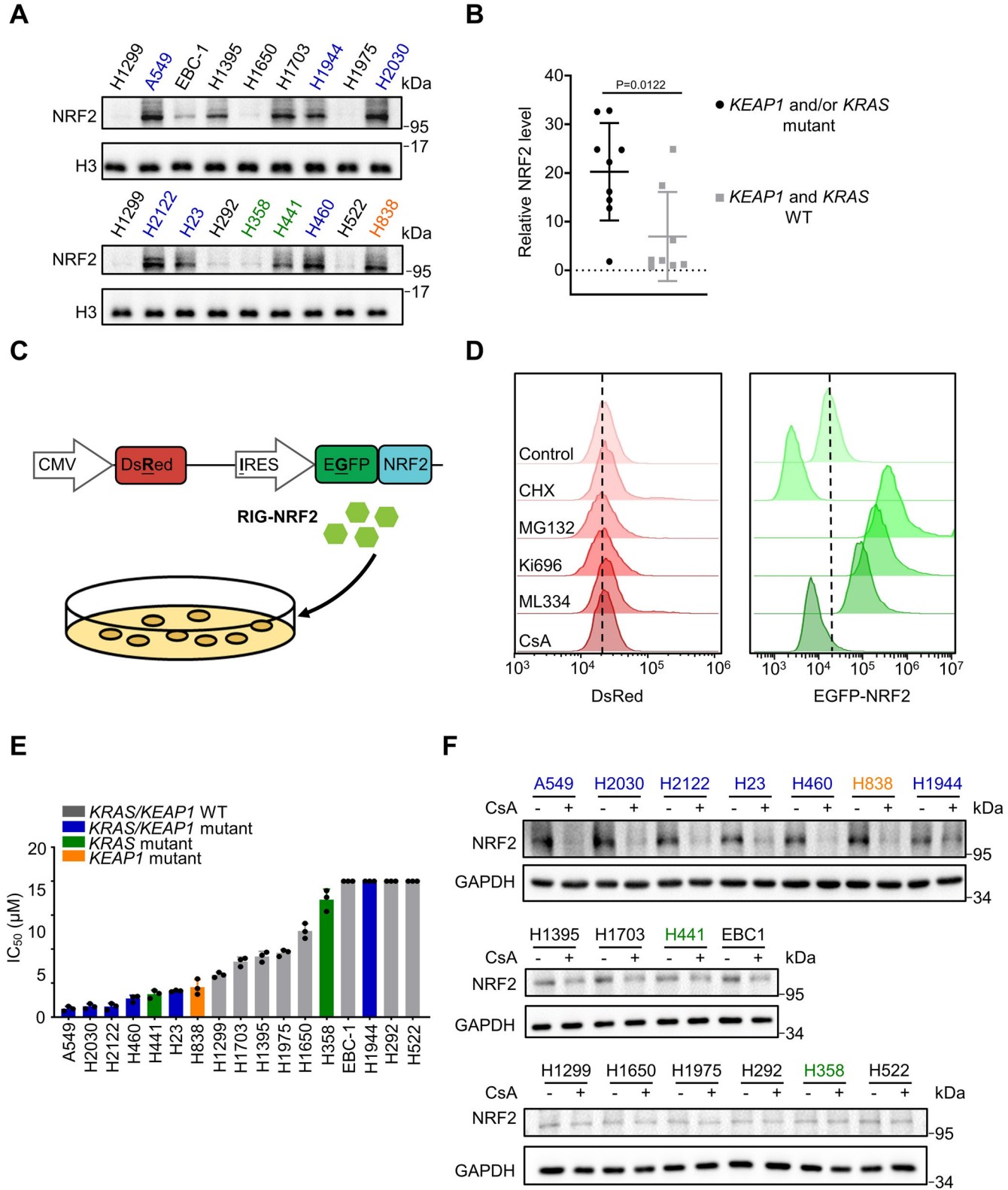

**Fig. 1 | Identification and validation of Cyclosporin A (CsA) as a chemical inducer of targeted NRF2 protein degradation. A** Representative immunoblot analysis of NRF2 protein levels in nuclear extracts of 17 NSCLC cell lines. **B** Dot plot showing the relative nuclear NRF2 protein levels in cell lines with *KEAP1* and/or *KRAS* co-mutations (*n* = 9 cell lines) and in cell lines with *KEAP1* and *KRAS* WT (*n* = 8 cell lines) related to Fig. 1A. **C** Overview of the DsRed-IRES-EGFP-NRF2 (RIG-NRF2) screen. CMV, CMV promoter; IRES, Internal ribosome entry site. **D** Flow cytometry analysis of DsRed and EGFP levels of A549-RIG-NRF2 cells treated with CHX (10 μM),

MG132 (10 μM), Ki696 (1 μM), ML334 (50 μM) and CsA (10 μM). The gating strategy are provided in Supplementary Fig. 13. **E** IC$_{50}$ values of CsA against a panel of NSCLC cell lines with distinct genetic profile. **F** Representative immunoblot analysis of NRF2 protein levels in a panel of NSCLC cells upon CsA treatment (10 μM). The results of panels (**A**, **D** and **F**) are representative of three independent experiments. **E** represents mean ± SD of three independent experiments. P value was analyzed using Two-tailed unpaired Student's t-test, *P* < 0.05 was considered statistically significant. Source data are provided as a Source Data file.

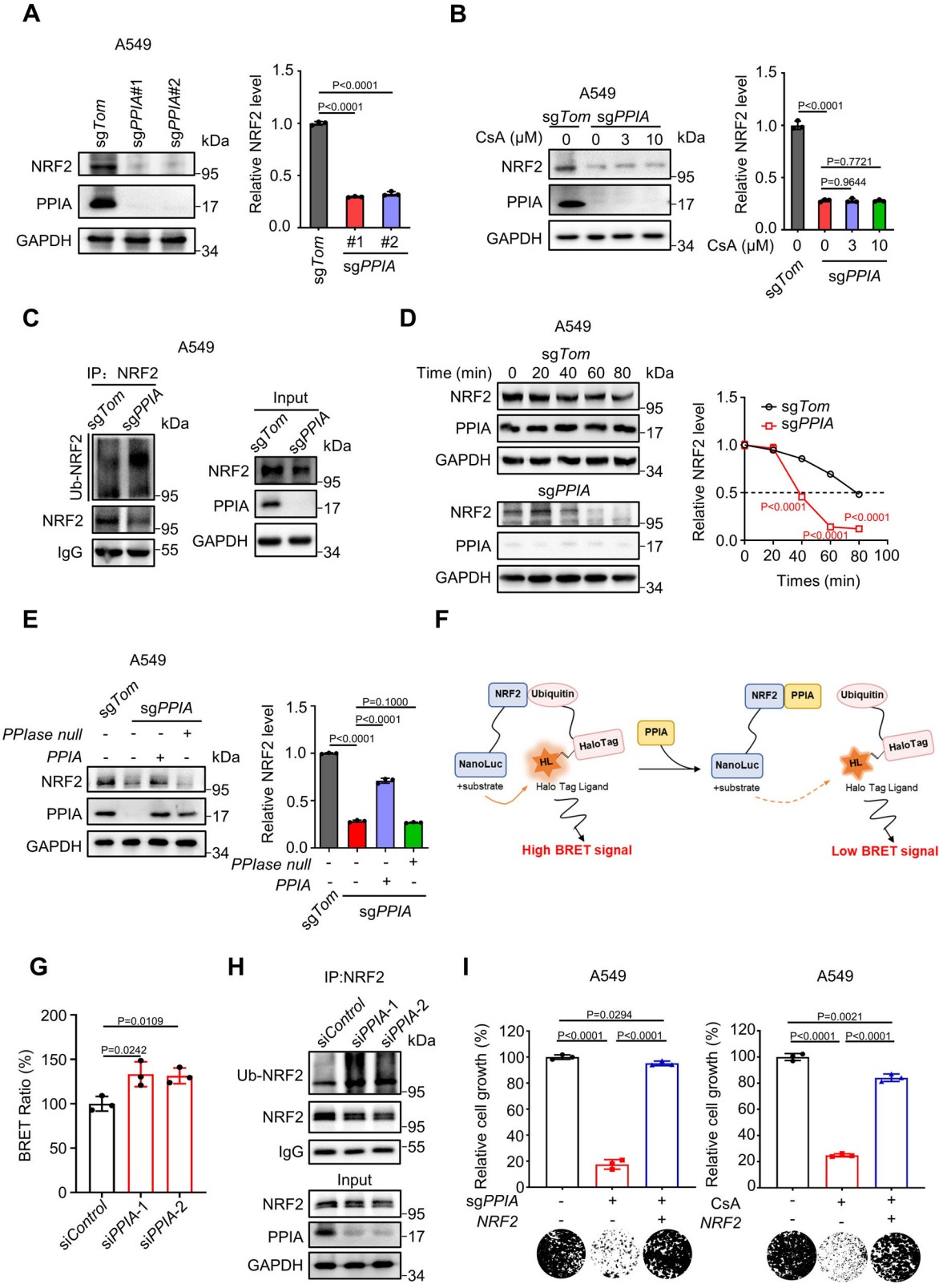

We next monitored NRF2 stability in WT and *PPIA*-KO A549 and H2030 cells by pre-treatment with cycloheximide (CHX) to block de novo protein synthesis. As a result, NRF2 protein half-life was much shorter in the *PPIA*-KO cells (A549, $t_{1/2} = 38.3 \pm 0.7$ min; H2030, $t_{1/2} = 30.1 \pm 1.3$ min) compared with the unedited control cells (A549, $t_{1/2} = 78.6 \pm 1.7$ min; H2030, $t_{1/2} = 61.2 \pm 2.8$ min) (Fig. 2D and Supplementary Fig. 3E). Moreover, we have additionally examined the

NRF2 stability in two NRF2-hyperactivated NSCLC cell lines (H460 and H2122). To do this, we knocked out (KO) the *PPIA* gene from H460 and H2122 using CRISPR/Cas9 technology. Consistently, NRF2 protein half-life was much shorter in the *PPIA*-KO cells (H460, $t_{1/2} = 44.5 \pm 1.4$ min; H2122, $t_{1/2} = 36.9 \pm 0.5$ min) compared with the unedited control cells (H460, $t_{1/2} > 80$ min; H2122, $t_{1/2} > 80$ min) (Supplementary Fig. 3F, G). Implicating PPIA's *cis-trans* isomerization activity on proline, we also

**Fig. 2 | Peptidylprolyl isomerase A (PPIA) is required for CsA-induced reduction of NRF2 levels and NSCLC cell proliferation. A** Representative immunoblot analysis of NRF2 protein levels in *PPIA*-WT or *PPIA*-KO A549 cells. Quantitative data was provided in the right panel. **B** Representative immunoblot analysis of NRF2 protein levels in *PPIA*-WT or *PPIA* KO A549 cells treated with CsA (0, 3, 10 μM) for 48 h. Quantitative data was provided in the right panel. **C** Immunoprecipitation of NRF2 followed by immunoblot analysis with anti-ubiquitin antibody detected the NRF2 ubiquitination in *PPIA*-WT and *PPIA*-KO A549 cells. **D** CHX chase assay of NRF2 protein stability in *PPIA*-WT or *PPIA*-KO A549 cells. Cells were treated with CHX (100 μg/mL) at the indicated time points and then subjected to immunoblot. Quantitative data was provided in the right panel. **E** Representative immunoblot results demonstrate that *PPIA* WT, but not *PPIA* catalytically dead *PPIA* variant (PPIA^{R55A&F60A}), can restore the NRF2 level in *PPIA*-KO A549 cells. Quantitative data

was provided in the right panel. **F** Schematic diagram of NanoBRET™ Ubiquitination Assay using the HaloTag®-Ubiquitin and NRF2-NanoLuc® fusion construct. **G** BRET signal of HEK293T cells transfected with HaloTag®-Ubiquitin and NRF2-Nanoluc® following si*PPIA* treatment or not. **H** Immunoprecipitation of NRF2 followed by immunoblot analysis with anti-ubiquitin antibody detected the NRF2 ubiquitination in HEK293T following si*PPIA* treatment or not. **I** Relative cell growth of A549 upon *PPIA* KO or CsA (10 μM) treatment in the presence or absence of *NRF2* overexpression. Quantitative results were shown in upper panel and representative colony image was presented in lower panel. **C** and **H** are representative of three independent experiments. (**A–E**), **G** and (**I**) represent mean ± SD of three independent experiments. *P* values were analyzed using Two-tailed unpaired Student's *t*-test, *P* < 0.05 was considered statistically significant. Source data are provided as a Source Data file.

overexpressed WT PPIA or a catalytically dead PPIA in the *PPIA*-KO background cells (R55A&F60A, retaining less than 1% enzymatic catalytic activity)[19]. Cells expressing WT PPIA, but not catalytically dead PPIA, displayed significantly elevated NRF2 levels (Fig. 2E and Supplementary Fig. 3H), indicating that catalytically competent PPIA is required for protecting NRF2 from degradation.

To further delineate the PPIA's protection effect on NRF2 from degradation by ubiquitin-proteasome system, we thus established a live-cell ubiquitin assay, NanoBRET™ Ubiquitination Assay, which can sensitively detect ubiquitination on a target protein (Promega). NanoBRET™ Ubiquitination Assay was performed using the HaloTag®-Ubiquitin and NRF2-NanoLuc® fusion proteins (Fig. 2F). As shown in Fig. 2G, a significantly increase in the NanoBRET™ signal was observed for NRF2/Ubiquitin in HEK293T upon si*PPIA* treatment, compared with the HEK293T cells upon si*Control* treatment. In line with these results, si*PPIA* treatment profoundly elevated the NRF2 ubiquitination as detected by immunoblotting (Fig. 2H). In contrast, si*PPIA* treatment failed to elevate the ubiquitination of NRF2 K7/R7 mutant (seven lysines in Neh2 domain of NRF2 required for NRF2 ubiquitination were all mutant into arginine)[20,21] (Supplementary Fig. 4A, B). Moreover, overexpression of WT PPIA, but not catalytically dead PPIA (R55A&F60A), displayed significantly reduced NanoBRET™ signal of NRF2/Ubiquitin and NRF2 ubiquitination by immunoblotting (Supplementary Fig. 4C, D). NRF2 ubiquitination is mediated by its negative regulator KEAP1, a component of an E3 ubiquitin ligase complex, upon binding[22]. We then asked whether PPIA affected the interaction of KEAP1 and NRF2. To this end, we performed co-immunoprecipitation analysis and we observed that overexpression of PPIA profoundly impaired the binding of KEAP1 to NRF2 (Supplementary Fig. 4E). These results suggest that PPIA can protect NRF2 from ubiquitination.

Next, we sought to establish a functional link between PPIA and NRF2. We found that firstly, NRF2-driven luciferase expression was markedly decreased in *PPIA*-KO A549 and H2030 cells and the inhibitory activities of CsA on ARE-reporter assay was abolished by *PPIA*-KO (Supplementary Fig. 4F). Secondly, CsA inhibits the mRNA expression of NRF2 target genes *HMOX1* and *NQO1* in a PPIA-dependent manner (Supplementary Fig. 4G). Thirdly, *PPIA*-KO or CsA treatment can significantly impair the colony formations in A549 and H2030 cells, but not in H1650 and H1975 cells with low basal NRF2 levels; overexpression of NRF2 in A549 and H2030 cells confers resistance to *PPIA*-KO and CsA treatment (Fig. 2I, Supplementary Fig. 4H,I). Finally, we employed A549 cells with inducible sh*Ctrl* or sh*PPIA* to generate tumors in mice. Doxycycline (20 mg/kg) was orally administrated to induce shRNA expression. Of note, *PPIA* knockdown strongly reduced growth of A549 xenograft tumors. In contrast, H1650 cells with inducible sh*Ctrl* or sh*PPIA* showed comparable tumor growth capacity (Supplementary Fig. 4J). Taken together, these results demonstrate that PPIA plays a vital role in protecting NRF2 protein from CsA-mediated degradation and consequently enhancing NRF2's transcriptional and proliferative functions in NSCLC cells.

## PPIA interacts with NRF2

Next, we investigated whether PPIA interacts with NRF2. Co-immunoprecipitation results revealed an interaction between PPIA and NRF2 using A549 cell lysates (Fig. 3A). Moreover, we found that PPIA's enzymatic activity is required for the PPIA-NRF2 interaction, since no interaction was observed for a catalytically dead PPIA variant (PPIA^{R55A&F60A}) (Fig. 3B). In addition, pull-down analysis using recombinant PPIA-conjugated beads showed that CsA profoundly blocked the interaction of PPIA and NRF2 (Fig. 3C). Collectively, these results indicate that PPIA can bind to NRF2.

## The complex crystal structure reveals that proline 174 (P174) of NRF2 is essential for PPIA binding

We then generated truncation variants to map the interaction of NRF2 and PPIA according to NRF2's functional NRF2-ECH homology (Neh) domains (Fig. 3D). Pull-down using recombinant PPIA-conjugated beads showed that the intrinsically disordered linker between the Neh4 and Neh5 domains (135-182aa), but not the Neh domains, is required for PPIA binding (Fig. 3E). Given the proline *cis–trans* isomerization activity of PPIA, we suspected possible involvement of proline residues in the PPIA/NRF2 interaction. Pursuing this, we mutated each of the five prolines in the flexible linker region (P145, P148, P154, P165, and P174) of NRF2 individually. Among these five prolines, P174A abolished the binding of NRF2 to PPIA in pull-down assay (Supplementary Fig. 5A). Correlated with this result, this NRF2 P174A mutant was less stable with a shorten half-life of 35.3 ± 1.4 min than NRF2 WT (t_{1/2} = 52.3 ± 3.7 min) (Fig. 3F). The NRF2 P174A mutant also elevated NanoBRET™ signal of NRF2/Ubiquitin and NRF2 ubiquitination by immunoblotting, suggesting a crucial role of PPIA in regulating NRF2 stability (Supplementary Fig. 5B, C).

We next sought to characterize the structural features of P174-dependent NRF2 and PPIA interaction. We synthesized a Proline-174-harboring hydrophobic peptide sequence (^{169}VAQVAPVD^{176}) located in the linker between the Neh4 and Neh5 domains and co-crystalized it with full-length PPIA using a hanging-drop apparatus. The structure of the complex was resolved with a resolution of 1.81 Å, and a final R_{free} of 0.2478 (Table 2, Supplementary Table 2). Overall, the structural data demonstrated that the VAPV fragment deeply embedded in the catalytic pocket of PPIA formed by R55, F60, M61, D66, F67, N101, and F113 (Fig. 3G, H). The guanidine of PPIA's R55 side chain directly interacts with the main chain of P174 of the NRF2 peptide. The amidogen and carboxyl of the PPIA's N102 main chain respectively form hydrogen bonds with A173 and V175 of the NRF2 peptide (Fig. 3H). Of note, P174 is buried in the catalytic cleft of PPIA in a *trans* conformation, which may be the result of the peptide bonds preceding and following P174, tightly anchoring P174 in the active pocket of PPIA. Intriguingly, the hydrophobic sequence motif ^{169}VAQVAPVD^{176} is highly conserved among primate species (Supplementary Fig. 5D), representing a unique PPIA-Binding-Motif (PBM).

Next, we compared the structural differences between PPIA-PBM and PPIA-CsA (PDB ID: 1CWA)[23]. The two structures present some

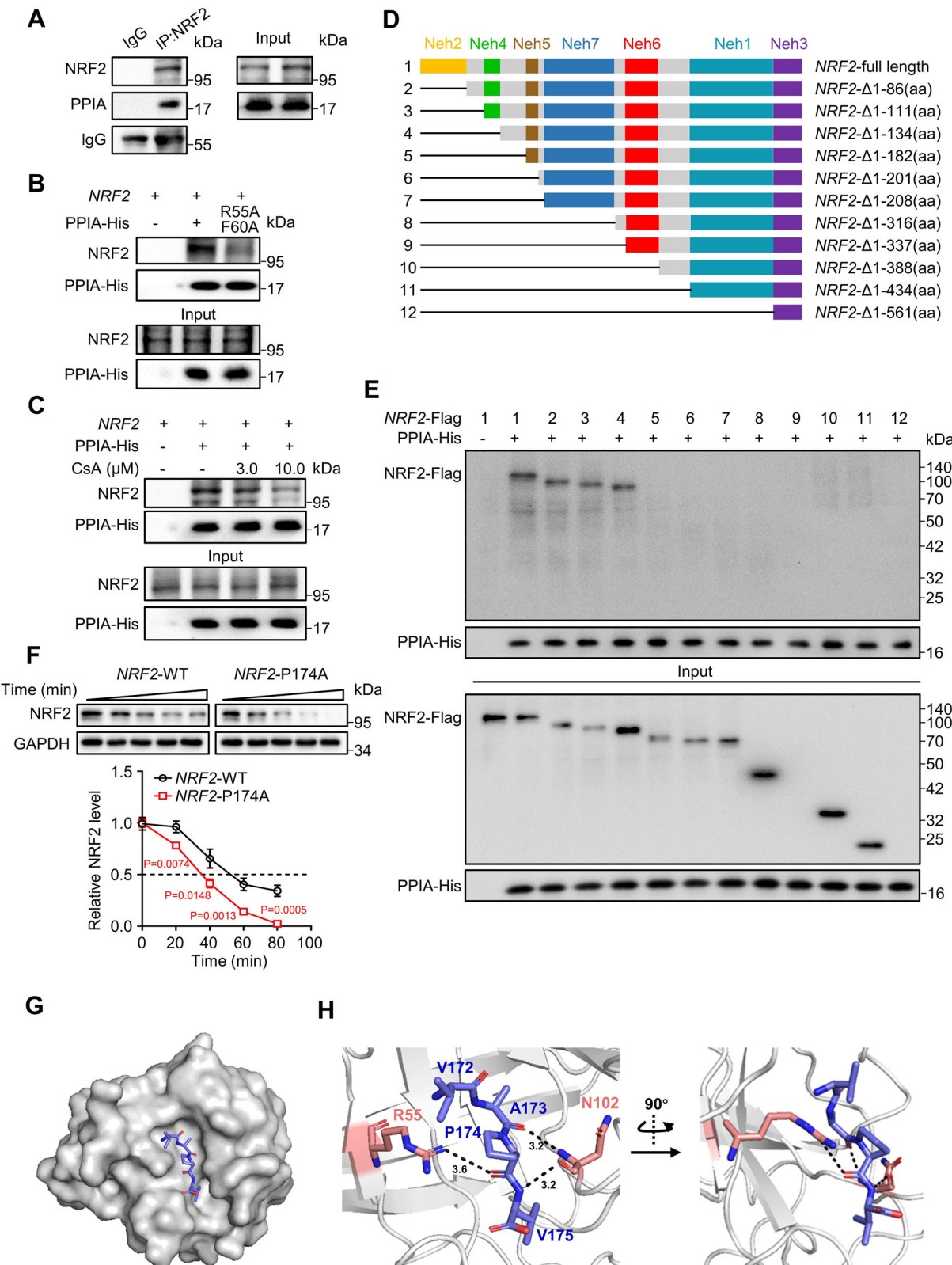

differences around the catalytic pocket, including the $\beta_3$ strand formed by residues H54 to I56, the $\beta_4$ strand formed by residues F60 to G64, and the $\alpha_2$ helix formed by residues E120 to D123 (Supplementary Fig. 5E). The most obvious variation is in the right-angle direction of guanidine in the R55 side chain (shifted by about 4.3 Å), leading to shifting of the $\beta_3$ strand's tail. A possible reason for this conformational change is strong interaction with VAPV that moves the R55 residue

towards P174 in PBM fragment. The VAPV peptide displayed a similar conformation at this location as residues 3 to 6 of CsA (Supplementary Fig. 5E), which can potentially explain CsA's disruption of the PPIA and NRF2 interaction.

We also employed a classic Nuclear Magnetic Resonance (NMR)-based approach to monitor the protein-peptide interaction. The $^{15}$N-edited heteronuclear single quantum coherence ($^{15}$N-HSQC)

**Fig. 3 | PPIA directly interacts with NRF2 and P174 of NRF2 is essential for PPIA binding. A** Co-immunoprecipitation analysis of endogenous proteins of PPIA and NRF2 using A549 cell lysates. **B** Pull-down assay of PPIA WT/NRF2 and PPIA^RSSA&F60A/NRF2. A549 cell lysate was incubated with PPIA WT or PPIA^RSSA&F60A pre-loaded beads, and NRF2 protein retained on the beads were detected by immunoblot. **C** The influence of CsA on the interaction between PPIA and NRF2. A549 cell lysate were incubated with PPIA pre-loaded beads in the presence of CsA (0, 3, 10 μM) and then subjected to pull-down assay. **D** Schematic diagram of various NRF2 truncation. **E** The binding of full-length and truncated NRF2 to PPIA as determined by pull-down assay. Various NRF2 truncations were overexpressed in HEK293T cells (as described in D). **F** CHX chase assay of NRF2 (WT or mutant P174A) protein stability. Cells were treated with CHX (100 μg/mL) at the indicated time points and then subjected to immunoblot (upper panel). Quantitative data was provided in the lower panel. **G** Crystal structure of PPIA in complex with NRF2 fragment PPIA-Binding Motif (PBM) ([169]VAQVAPVD[176]). PBM peptide is deeply embedded into catalytic pocket of PPIA presented as gray surface. [172]VAPV[175] residues of NRF2 peptide are displayed as slate stick. **H** Expansion of the catalytic pocket of PPIA in complex with NRF2 fragment PBM. PPIA is shown in gray cartoon and the interacted residues are displayed as salmon sticks. NRF2 PBM fragment is presented as slate sticks. The black dashed line denotes the hydrogen contact. The results of panels (**A–C**, **E**, **F**) are representative of three independent experiments. **F** represents mean ± SD of three independent experiments. P values were analyzed using Two-tailed unpaired Student's t-test, P < 0.05 was considered statistically significant. Source data are provided as a Source Data file.

spectrum revealed that the addition of the PBM peptide solution resulted in pronounced chemical shift perturbations. In line with the complex structure of PPIA and PBM fragment, a residue-specific NMR spectroscopy showed that C52, G59, D66, F67, and F113—all located in the catalytic pocket of PPIA—as the most affected residues (Supplementary Fig. 5F, G). Together, these results demonstrate that NRF2 via its unique PBM motif directly binds to PPIA and P174 is vital for this binding and maintaining NRF2 stability by PPIA.

## PPIA inhibition impairs glutamine metabolism in NSCLC cells with NRF2 hyperactivation

Previous studies established that abnormal NRF2 activation promotes glutamine addiction in NSCLC[24,25]. We therefore evaluated the effect of PPIA inhibition on glutamine metabolism in A549 and H2030 cells. We found that glutamine supplementation significantly enhanced cell growth, whereas the growth-promoting effects of this supplementation were impeded upon *PPIA* KO and upon CsA treatment of both cell lines (Fig. 4A, Supplementary Fig. 7A). Glutamine is converted to glutamate by glutaminase and further oxidized by tricarboxylic acid (TCA) cycle in mitochondrial[26]. A549 and H2030 cells were loaded with minimal medium in which glutamate was used as sole carbon source. These two cell lines were subjected to oxygen consumption rate (OCR) assay via a Seahorse XFe96 analyzer. We found that *PPIA* KO and CsA treatment results in reduced basal and maximal OCR, suggesting impaired glutamine metabolism in mitochondria of both cell lines (Fig. 4B, Supplementary Fig. 7B). We further performed the glutamine metabolism tracing using $^{13}C_5$-glutamine and LC-MS/MS (Supplementary Fig. 6A). As shown in Supplementary Fig. 6B, *PPIA* KO can significantly decrease the flux of glutamine-derived TCA metabolites, including glutamate, α-ketoglutarate, succinate, fumarate, and malate. Consistent with these observations, the supplementation of glutamate, α-ketoglutarate, succinate, and fumarate can significantly rescue the cell viability upon *PPIA* KO (Supplementary Fig. 6C).

We also examined whether PPIA affects glutamine uptake/transport by using Q-PCR to assess the transcription levels of human glutamine transporters (Fig. 4C). The largest decrease in transcription in *PPIA*-KO cells was the transcripts of the *SLC1A5*, a high-affinity glutamine transporter for maintaining influx of glutamine. We also observed a significant decrease in both the mRNA and protein levels of SLC1A5 upon *PPIA* KO and upon CsA treatment in both A549 and H2030 cells. Of note, overexpression of *NRF2* reversed SLC1A5 expression loss (Fig. 4D–G, Supplementary Fig. 7C–F).

We next investigated the effects of PPIA inhibition on cellular glutamine content. As anticipated, *PPIA* KO and CsA treatment resulted in 68.42 ± 1.25% and 67.09 ± 4.10% reductions of intracellular glutamine content in A549 cells, respectively. Whereas, *NRF2* overexpression effectively restored glutamine level (Fig. 4H). The similar results were also observed in H2030 cells (Supplementary Fig. 7G). To assess the functional role of SLC1A5 in glutamine transport, we overexpressed *SLC1A5* in A549 and H2030 cells. As expected, we found that *SLC1A5* overexpression rescued the reduction in glutamine content upon *PPIA* KO and upon CsA treatment in these two cell lines (Fig. 4H and Supplementary Fig. 7G). We also observed that *SLC1A5* overexpression effectively rescued the cell growth inhibition in both A549 and H2030 cells upon *PPIA* KO and upon CsA treatment (Fig. 4I, Supplementary Fig. 7H). Taken together, these results suggest that PPIA inhibition may impede glutamine metabolism in a SLC1A5-dependent manner in NRF2-hyperactivated NSCLC cells.

## NRF2 activates SLC1A5 transcription through its downstream gene *KLF5*

Immunoblotting against NRF2 and SLC1A5 in a panel of 17 NSCLC cell lines showed that these two proteins were commonly co-expressed (Supplementary Fig. 8A; Pearson R = 0.7271, P = 0.0009). RNA-sequencing data of NSCLC tissues derived from GEO database (GSE8894) also indicated a positive correlation between *NRF2* and *SLC1A5* (Supplementary Fig. 8B; Pearson R = 0.5197, P < 0.0001)[27]. We then examined the possibility that *SLC1A5* may be a direct target gene of the NRF2 transcription factor by generating a luciferase reporter construct containing the *SLC1A5* promoter (ranging from -2000 bp to +100 bp). Overexpression of *NRF2* increased the promoter activity of *SLC1A5*, doing so in a dose-dependent manner (Supplementary Fig. 8C). To map the transcription activation region, we assessed a series of truncated *SLC1A5* promoters and ultimately identified the binding region from -500 bp to -10 bp (Supplementary Fig. 8D). However, analysis using the JASPAR database[28] did not detect any NRF2 binding sequence (ARE element) within this region, strongly suggesting that NRF2 per se is not responsible for the observed transcriptional activation of *SLC1A5*.

We therefore suspected an indirect regulation mechanism, so we looked for transcription factors common to i) the NRF2-regulated genes identified based on chromatin immunoprecipitation (ChIP)-Seq

**Table 2 | X-ray data collection statistics**

| Data statistics | PPIA/NRF2-PBM complex |
|---|---|
| Data collection date | 2020/12/25 |
| Wavelength | 1.0 Å |
| Detector | PILATUS 6 M |
| PDB code | 8HZ8 |
| Space group | $P2_12_12_1$ |
| a, b, c (Å) | 42.490, 52.326, 88.501 |
| α, β, γ (°) | 90, 90, 90 |
| Resolution range (Å) | 26.446–1.81 (1.95–1.81) |
| Reflections | 16105 |
| I/σ | 1.31 |
| CC1/2 | 0.996 |
| Wilson B-factor | 29.7 |
| Completeness (%) | 99.9 (25.70-1.81) |
| Redundancy | 19.5 |

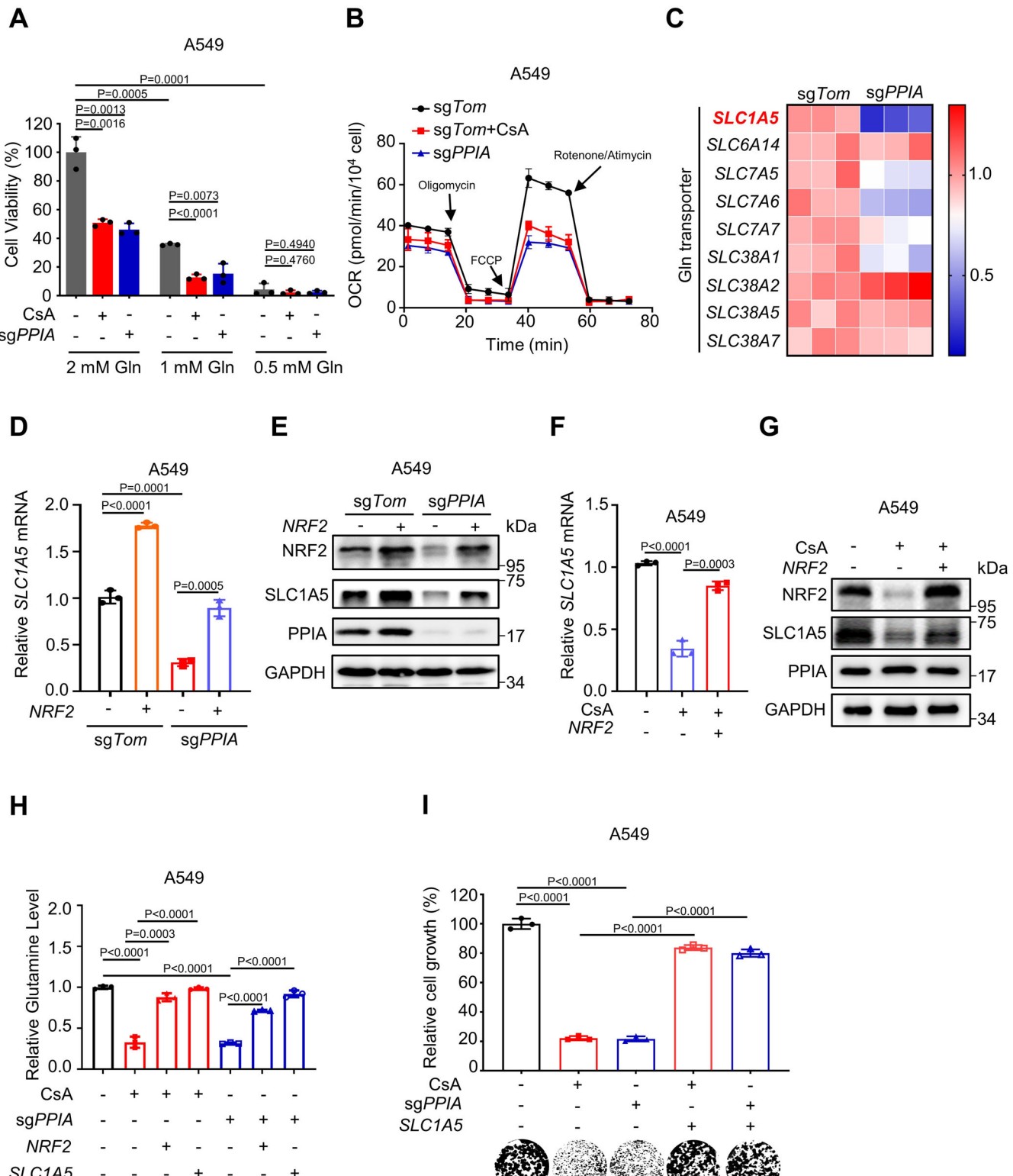

**Fig. 4 | PPIA inhibition impairs glutamine metabolism in NSCLC cells with NRF2 hyperactivation. A** The cell viability of A549 cells upon CsA (10 μM) treatment or upon *PPIA* KO in the presence of glutamine (2, 1, 0.5 mM) was measured by MTT assay. Gln, glutamine. **B** Oxygen consumption rate (OCR) plotted over time in A549 cells following CsA (10 μM) treatment or *PPIA* KO. **C** Heatmap showing the mRNA levels of different glutamine membrane transporters in *PPIA*-WT or *PPIA*-KO A549 cells. **D-E** Q-PCR **D** and immunoblot **E** analysis of SLC1A5 in *PPIA*-WT or *PPIA*-KO A549 cells following *NRF2* overexpression. **F-G** Q-PCR **F** and immunoblot **G** results of SLC1A5 in A549 cells treated with CsA (10 μM for 48 h) with or without *NRF2* overexpression. **H** Relative glutamine levels in A549 cells upon CsA (10 μM) treatment or upon *PPIA* KO, following *SLC1A5* or *NRF2* overexpression. **I** A549 cells upon CsA (10 μM) treatment or upon *PPIA* KO following *SLC1A5* overexpression were subjected to colony formation. Quantitative results were shown in upper panel and representative colony image was presented in lower panel. The results of panels (**C**, **E**, **G**) are representative of three independent experiments. **A**, **B**, **D**, **F**, **H** and **I** represent mean ± SD of three independent experiments. P values were analyzed using Two-tailed unpaired Student's t-test, P < 0.05 was considered statistically significant. Source data are provided as a Source Data file.

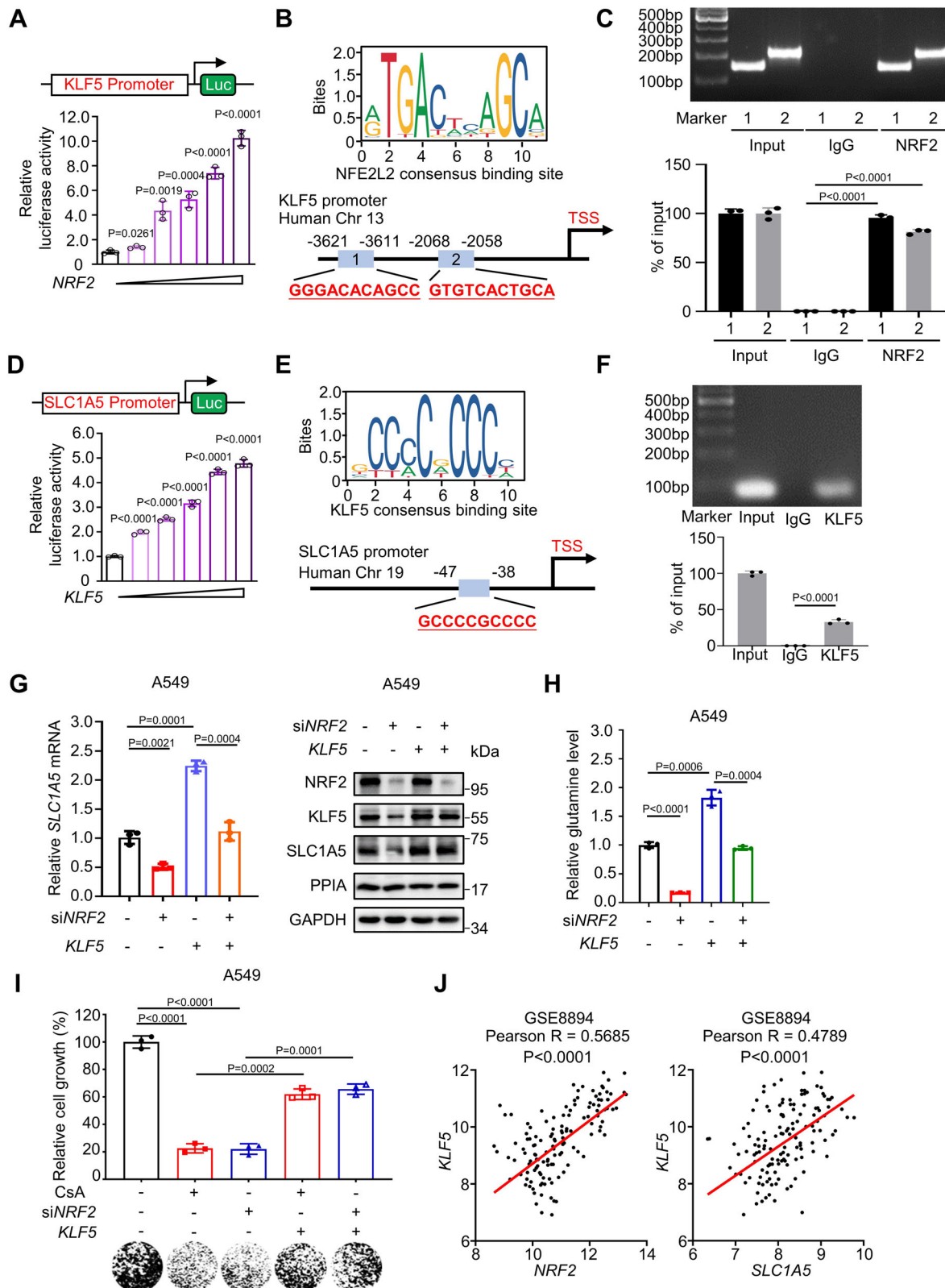

data[29] and ii) predicted transcription factor binding to the promoter of *SLC1A5* in the GeneCards database[30] (Supplementary Fig. 8E). Among the 14 such genes, Kruppel Like Factor 5 (*KLF5*) exhibited the highest predicted JASPAR score (Supplementary Fig. 8F). Firstly, we examine whether *KLF5* is a target gene of the NRF2. We found that NRF2 enhanced *KLF5* transcription using a luciferase reporter vector containing the *KLF5* promoter (ranging from -4000 bp to 0 bp) (Fig. 5A).

Two potential NRF2 binding sequences (ARE element) within the KLF5 promoter (GGGACACAGCC, -3621 to -3611 bp and GTGTCACTGCA, -2068 to -2058 bp) (Fig. 5B) were predicted by JASPAR and confirmed by ChIP-PCR (Fig. 5C). Subsequently, we asked whether KLF5 induce *SLC1A5* transcription. Luciferase reporter (the -500 bp to -10 bp region) analysis demonstrated that KLF5 was capable of inducing *SLC1A5* transcription (Fig. 5D). ChIP-PCR results demonstrated that KLF5 binds

**Fig. 5 | NRF2 activates *SLC1A5* transcription via its downstream gene *KLF5*.**
**A** pGL3-Luc vector containing human *KLF5* promoter (-4,000 to 0 bp) and pCDNA3.1-Flag-NRF2 were co-transfected into HEK293T cells with a ratio of 1:0.5 to 1:8 and luciferase activity was determined. **B** Analysis of NRF2 consensus motif enrichment in the *KLF5* promoter (-4,000 to 0 bp) predicted by JASPAR database. Matched consensus motifs are shown in schematic. **C** ChIP-PCR analysis of the enrichment of NRF2 at the promoter region of *KLF5* in A549 cells. **D** pGL3-Luc vector containing human *SLC1A5* promoter (-500 to +10 bp) and pCDNA3.1-Flag-KLF5 were co-transfected into HEK293T cells with a ratio of 1:0.5 to 1:8 and luci-ferase activity was determined. **E** Analysis of KLF5 consensus motif enrichment in the *SLC1A5* promoter predicted by JASPAR database. Matched consensus sequen-ces are in bold. **F** ChIP-PCR analysis of the enrichment of KLF5 at the promoter region of *SLC1A5* in A549 cells. **G** Q-PCR and immunoblot results of SLC1A5 in A549 cells treated with si*NRF2* in the presence or absence of *KLF5* overexpression. **H** Relative glutamine level in A549 cells following si*NRF2* treatment in the presence or absence of *KLF5* overexpression. **I** Colony formation of A549 cells treated with CsA or si*NRF2* in the presence or absence of *KLF5* overexpression. Quantitative results were shown in upper panel and representative colony image was presented in lower panel. **J** Correlation analysis of *KLF5/NRF2* or *KLF5/SLC1A5* gene expression in clinical NSCLC tumor samples (*n* = 138 samples). The data are derived from public dataset (GSE8894) and analyzed in PrognoScan. The results of panels (**C**, **F**, **G**) are representative of three independent experiments.
**A**, **C**, **D**, **F**, **G–I** represent mean ± SD of three independent experiments. P values were analyzed using Two-tailed unpaired Student's t-test, P < 0.05 was considered statistically significant. Source data are provided as a Source Data file.

to a specific motif (GCCCCGCCCC, -47 to -38 bp) within the *SLC1A5* promoter (Fig. 5E, F).

Next, we observed that *NRF2* overexpression increased the mRNA levels of KLF5, while si*NRF2*-mediated knockdown significantly reduced KLF5 accumulation (Fig. 5G and Supplementary Fig. 9A, B). Additionally, we found that the effects of *NRF2* knockdown in A549 and H2030 cells—including decreased mRNA level and protein accumulation of SLC1A5 and decreased intracellular glutamine content—were restored upon *KLF5* overexpression (Fig. 5G, H, Supplementary Fig. 9B, C). Moreover, *KLF5* knockdown reduced the intracellular glutamine level and the colony formation capacity of A549 and H2030 cells, which were restored following *SLC1A5* overexpression (Supplementary Fig. 9D,E). By contrast, *KLF5* overexpression rescued the growth capacity of A549 and H2030 cells upon CsA treatment and NRF2 knockdown (Fig. 5I, Supplementary Fig. 9F). We further examined the co-expression rela-tionship of NRF2, KLF5 and SLC1A5 in NSCLC. Quantitation of the immunoblotting showed protein expression of KLF5 is positively cor-related with that of NRF2 or SLC1A5 in NSCLC cell lines (Supplementary Fig. 8A). Additionally, analysis of the human NSCLC dataset in the GEO database (GSE8894)[27] suggested that the mRNA expression of KLF5 is positively associated with that of NRF2 or SLC1A5 (Fig. 5J). These results collectively demonstrate that NRF2-KLF5 signaling promotes the tran-scription activation of glutamine transporter SLC1A5 in NSCLC cells.

## Co-targeting of PPIA and glutaminase potently inhibits tumor growth of NRF2-hyperactivated NSCLC

Glutamine metabolism is explored therapeutically as a new approach for the treatment of many types of cancers, including NSCLC[31]. Glu-tamine is transported into cells through plasma membrane transpor-ters (e.g., SLC1A5) and is subsequently converted to glutamate by glutaminase to fuel cancer cell growth[32]. We next evaluated the anti-proliferation effects of a combination therapy comprising CsA and glutaminase inhibitor CB-839 against A549 and H2030. Co-treatment with CsA and CB-839 caused a pronounced reduction in the viability of both A549 and H2030 cells, with the co-treatment outperforming either of the single agents (Supplementary Fig. 10A). Combining CsA with CB-839 demonstrated a synergistic anti-proliferation effect with combination index (CI) values less than 1 at varying concentrations (Supplementary Fig. 10A). Notably, overexpression of KEAP1 pro-foundly reduced NRF2 protein levels and conferred resistance to CsA and CB839 combination treatment in both A549 and H2030 cells (Supplementary Fig. 10B, C). H1650 and H1975 cells with low NRF2 expression are insensitive to CsA/CB-839 co-treatment (Supplemen-tary Fig. 10D). Note that a glutamine deprivation assay indicated that the anti-proliferation effect from the CsA and CB-839 drug combina-tion was dependent on the presence of glutamine (Fig. 6A and Sup-plementary Fig. 10E). We also conducted 3D colony formation analysis with A549 and H2030 cells and observed a significant loss of colony formation capacity upon the CsA and CB-839 combination treatment, which again exceeded either of the single agent treatments (Fig. 6B and Supplementary Fig. 10F).

We next assessed the potency of combined CsA and CB-839 in vivo. Luciferase-labeled A549 (A549-Luc) cells were injected into the tail vein of nude mice, and tumor progression was monitored using an IVIS imaging system. Mice were administered with CsA (20 mg/kg, i.p., every three days), CB-839 (150 mg/kg, p.o., twice daily), or their combination. 3D bioluminescence imaging showed that the vehicle control animals developed massive tumors in their lungs. While both the CsA and CB-839 monotherapies reduced the tumor burden compared to the vehicle control, it was obvious that the combination therapy conferred superior anti-tumor effects (Fig. 6C). Consistently, analysis of isolated lung tis-sues also showed that the CsA and CB-839 combination therapy resulted in significantly fewer tumor foci compared to the vehicle control and compared to both monotherapies (Fig. 6D). It was also highly notable that the CsA and CB-839 drug combination significantly extended the survival of tumor-bearing animals (Fig. 6E; *P* < 0.0001). Note that there was no obvious change in mouse body weight in any of the groups, suggesting that these treatments were well tolerated by the animals (Supplementary Fig. 11A). Moreover, considering the potential immu-nosuppressive activity of CsA, we also examined the efficacy of CsA and CB-839 drug combination using humanized NCG mice with A549-implanted tumors. Of note, CsA and CB-839 combination therapy have similar anti-tumor activity in immune-competent humanized NCG mice, compared with immunodeficient nude mice (Supplementary Fig. 11B).

Finally, the in vivo efficacy of CsA and CB-839 drug combination was evaluated in clinically relevant patient-derived xenograft (PDX) models. We employed five NSCLC PDX models with distinct geno-types: the MT-101 and MT-102 models harbor concurrent mutant *KEAP1/KRAS*, while the WT-201, WT-202 and WT-203 models exhibit *KEAP1/KRAS* WT (Supplementary Table 3). Consistent with the co-occurring *KEAP1/KRAS* genotype, we detected an obvious increase in the NRF2 protein level in the MT-101 tumors compared to WT-201 tumors (Supplementary Fig. 11C). For the MT-101 PDX model, the CsA and CB-839 monotherapies each moderately significantly reduced the tumor volumes, with inhibition rates of 29.83 ± 9.50% and 49.41 ± 5.91%, respectively, compared to the vehicle control. The CsA and CB-839 combination therapy performed better, achieving an inhibition rate of 75.17 ± 4.20% (Fig. 6F and Supplementary Fig. 11D). We also examined the anti-tumor synergy of CsA and CB-839 in MT-101 PDX model using a Q value method[33–35]. The Q value of CsA plus CB-839 was 1.165 (Q ≥ 1.15 indicates synergism), suggesting a syner-gistic effect between CB-839 and CsA. Of note, we observed a similar anti-tumor potency of CsA and CB-839 combination therapy in MT-102 PDX model (Supplementary Fig. 11F). By contrast, for *KEAP1/KRAS* WT PDX models we detected no significant differences in tumor growth between the vehicle control and any of the three treatments (CsA, CB-839, or their combination) (Fig. 6G, Supple-mentary Fig. 11E, G and H). These findings support that the observed therapeutic benefits of the CsA and CB-839 combination can be attributed to NRF2 hyperactivation.

Consistent with our earlier observations about the NRF2/KLF5/SLC1A5 pathway in cultured cells, immunoblotting of the MT-101

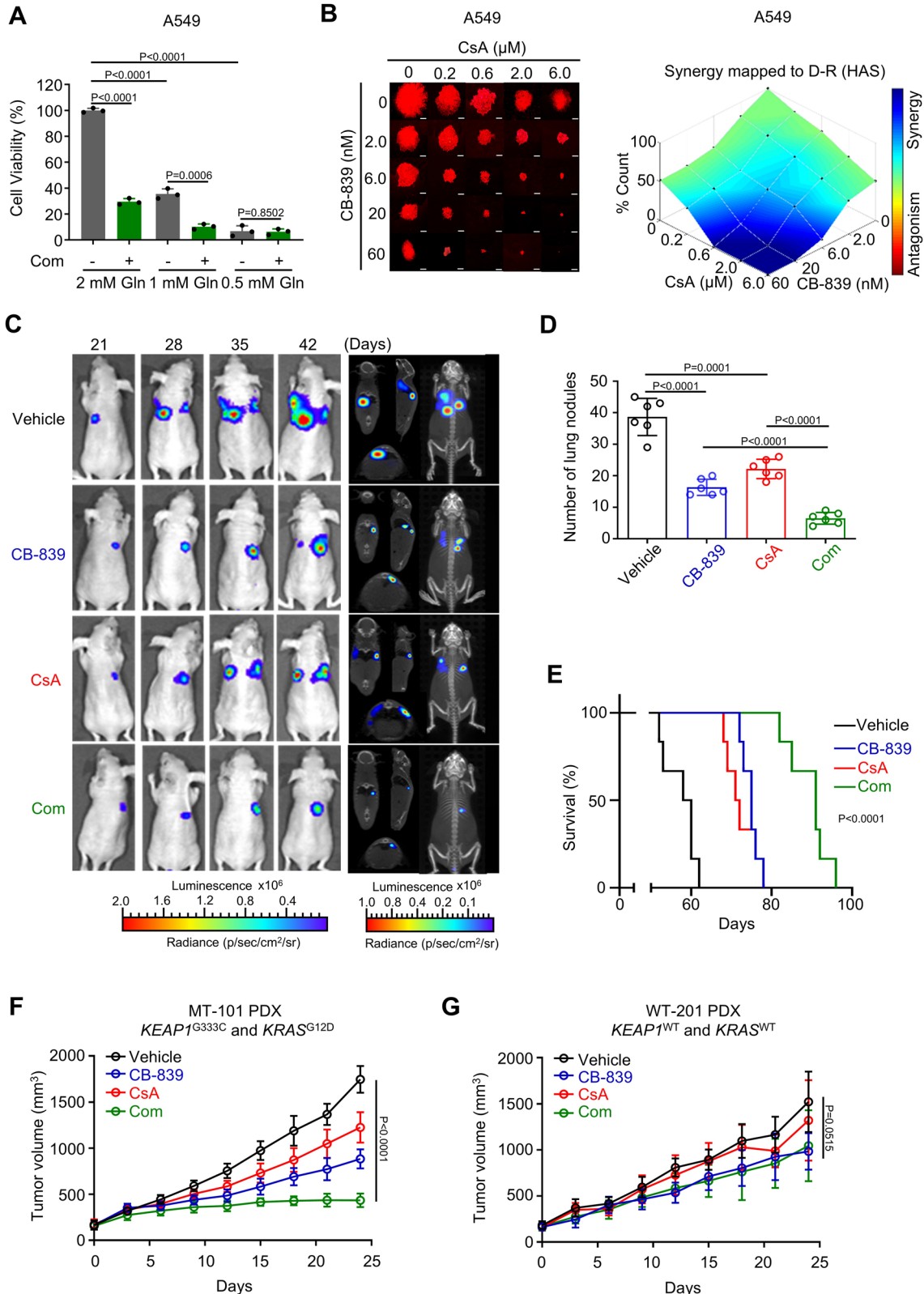

tumor tissues showed that the accumulation of NRF2, KLF5, and SLC1A5 was obviously reduced upon CsA treatment and upon the CsA and CB-839 combination treatment (Supplementary Fig. 11I). In contrast, CsA alone or combined with CB-839 have minor effects on NRF2, KLF5, and SLC1A5 proteins derived from WT-201 tumor samples (Supplementary Fig. 11J). Collectively, these results demonstrate that simultaneously targeting of PPIA and glutaminase represents a

promising strategy for the treatment of NSCLC patients harboring NRF2 hyperactivation.

## Elevated PPIA and NRF2 accumulation is positively associated with poor prognosis in NSCLC patients

To explore the clinical significance of PPIA and NRF2 in NSCLC, we examined their accumulation using a tumor tissue microarray

**Fig. 6 | Co-targeting of PPIA and glutaminase potently inhibits tumor growth of NRF2-hyperactivated NSCLC. A** Cell viability of A549 cells treated with CsA (8 μM) and CB-839 (0.08 μM) combination (Com) in presence of 2, 1, 0.5 mM glutamine (Gln). **B** 3D clonogenic assay of A549-mCherry cells treated with vehicle, CsA, CB-839, or their combination (n = 3 independent experiments). After growing for 2 weeks, formed clones were quantitated by using Image J. The synergistic anti-proliferative effect was evaluated by Combenefit. Blue indicates synergy, while red indicates antagonism between drugs. **C** Lung orthotopic model was established by tail-vein injection of A549-Luc cells. Mice were divided into four groups (n = 6 mice per group): vehicle control (p.o. twice daily), CB-839 (150 mg/kg, p.o. twice daily), CsA (20 mg/kg, i.p. every 3 days), and CB-839/CsA combination (150 mg/kg, p.o. twice daily; 20 mg/kg, i.p. every 3 days). Bioluminescence imaging was performed every 7 days using a PerkinElmer IVIS Spectrum CT system. **D** Statistics of lung

metastatic nodules in the lung orthotopic model with different treatments (n = 6 mice per group). **E** Kaplan–Meier survival analysis of different groups in lung orthotopic model formed by tail-vein injection of A549-luc cells (n = 6 mice per group). **F, G** Tumor growth in NCG mice bearing MT−101 **F** or WT-201 **G** PDX xenograft treated with the vehicle (p.o. twice daily), CB-839 (150 mg/kg, p.o. twice daily), CsA (20 mg/kg, i.p. every 3 days), and CB-839/CsA combination (n = 6 mice per group). MT−101 PDX harbors concurrent *KEAP1* and *KRAS* mutations (*KEAP1*[G333C] and KRAS[G12D]). WT-201 PDX has *KEAP1*[WT] and *KRAS*[WT]. **A** represents mean ± SD of three independent experiments. P values were analyzed using Two-tailed unpaired Student's t-test for **A** and **D**, using Log-rank (Mantel-Cox) test for **E**, and using One-way ANOVA for (**F, G**). P < 0.05 was considered statistically significant. Source data are provided as a Source Data file.

comprising specimens of multiple diagnosed clinical stages. According to the immunohistochemical staining of paired tumor/normal tissues, PPIA and NRF2 expression were dramatically higher in NSCLC specimens than in adjacent normal lung tissues (Fig. 7A). Moreover, approximately 86.48% of the PPIA-high samples showed high expression level of NRF2, whereas 65.62% of clinical NSCLC specimens with low level of PPIA exhibited low NRF2 expression (Table 3, P < 0.001). Quantitation of the immunohistochemical staining also showed highly correlation of PPIA and NRF2 in NSCLC tissues (Fig. 7B, Pearson R = 0.6996, P < 0.0001). Further Kaplan-Meier survival analysis revealed that patients with high PPIA or high NRF2 expression had poorer prognosis than those with low PPIA or low NRF2 expression (Fig. 7C, D; Logrank P < 0.0001 for PPIA and Logrank P = 0.0117 for NRF2, respectively).

We also analyzed the prognostic significance of PPIA and NRF2 using the Kaplan-Meier plotter database[36]. In line with the results of the tissue microarray, high mRNA expression of *PPIA* and *NRF2* were associated with reduced overall survival time as well (Fig. 7E-7F; Logrank P < 0.0001 for *PPIA* and Logrank P < 0.0001 for *NRF2*, respectively). Additionally, we noted that NSCLC patients with higher *KLF5* and *SLC1A5* mRNA expression predict poor prognosis (Fig. 7G-7H; Logrank P = 0.0017 for *KLF5* and Logrank P < 0.0001 for *SLC1A5*, respectively). Together, these data demonstrate that PPIA and NRF2 are positively associated with malignant progression and poor prognosis of NSCLC.

## Discussion

More than 5% of the cancer patients were diagnosed as NRF2-mutant/hyperactivated cancers, as revealed by TCGA Pan-Cancer analysis, with significant enrichment in lung malignancy[37–39]. Accumulating evidence has established that NRF2 hyperactivation as a driver of cancer progression, metastasis, and therapy resistance[4]. Although the therapeutic appeal of NRF2 inhibition, the transcription factor is a tricky target. Research efforts have focused on alternative strategies aimed at exploring NRF2-related vulnerabilities. In the current study, we demonstrate that PPIA is essential for NRF2 stability and is a druggable vulnerability in NRF2-hyperactivated NSCLC.

NRF2 is a fundamental regulator of homeostatic milieu in response to diverse stress stimuli in most metazoans[40,41]. NRF2 protein in the cell was tightly controlled by ubiquitin proteasomal pathway[42]. It has previously been reported that NRF2-ECH homology (Neh) domains, such as Neh2, are critical determinants of NRF2 protein stability[43]. Herein, we demonstrated the existence of a specific primate-conserved sequence PBM located in the linker between the Neh4 and Neh5 domains, which is indispensable for governing NRF2 protein stability. PPIA binds to the P174-containing PBM fragment of NRF2 and blocks the access of ubiquitin/KEAP1 to NRF2, thus inhibiting ubiquitin-proteasome dependent degradation. P174A mutant fails to bind to PPIA with a shorten half-life. Our high-resolution co-crystal structure reveals that the *trans* P174-containing hydrophobic fragment is embedded in the catalytic pocket of PPIA. These findings open a new

horizon of the complex molecular mechanisms that govern NRF2 protein stability in health and disease. Intriguingly, TCGA analysis indicated that the interaction interface of PPIA and NRF2 does not harbor mutations in lung and other cancers (Supplementary Fig. 12A, B). The biochemical and structural insights into the regulation of transcription factor NRF2 by PPIA may offer new therapeutic approaches for clinical interventions through disturbing the protein-protein interaction[44].

PPIA exhibits peptidylprolyl cis-trans isomerase activity, which modulates substrate protein folding/stability and cell signaling[45]. PPIA has been identified as a dysfunctional protein in a variety of human cancers[46–48]. PPIA promotes the colonization and proliferation of multiple myeloma cells via binding to CD147[49]. A recent study further identified PPIA as an attractive target for the treatment of resistant multiple myeloma[50]. Charalampos G Kalodimos group have reported that PPIA binds to and enhances activation of CrkII, which stimulate breast cancer cell migration[51]. Yet, the molecular mechanisms underlying PPIA action and the signaling pathways in NSCLC are still poorly understood. In this study, we discovered that PPIA modulates NRF2 stability, and its prolyl cis-trans isomerase activity renders cancer progression of NRF2-hyperactivated NSCLC. However, we cannot completely exclude the implication of other PPIA substrate proteins for PPIA's oncogenic properties in NSCLC cells, despite we found that two known substrate proteins CrkII and TARDBP are not affected by PPIA inhibition/KO (Supplementary Fig. 12C, D). Of note, tumor tissue microarray and bioinformatics analysis indicated that PPIA and NRF2 are vastly upregulated in patients with NSCLC. Our findings highlight a potential mechanism through which PPIA regulate NRF2 stability and reveal an appealing translational value of targeting PPIA in NRF2-hyperavtivated malignancies. In addition, PPIA is known to regulate transcription factor NFAT, which plays important roles in cancer progression[52]. Our findings raise the possibility that PPIA may act as a molecule node for the cross-talk between NRF2 and NFAT signaling in cancer. Further molecular mechanistic understanding of the reciprocal interactions between NRF2 and NFAT pathway may inspire novel biological insights and therapeutic regimens aimed at targeting tumor progression.

The FDA-approved macrocyclic peptide CsA impairs the binding of NRF2 to PPIA and triggers NRF2 degradation via ubiquitin-proteasome system. CsA represents a chemical inducer of NRF2 degradation. As a result, NRF2-hyperactivated NSCLC cell lines harboring KEAP1 and/or KRAS mutations, are highly sensitive to CsA treatment. And CsA significantly impairs the growth of NRF2-hyperactivated tumors in the clinically relevant PDX models. CsA is a widely used immunomodulatory drug, included in the World Health Organization (WHO) Model List of Essential Medicines (https://list.essentialmeds.org/?query=ciclosporin), with well-documented effectiveness and safety profile[53,54]. Clinically, CsA has been investigated for the possibility of being repurposed for the treatment of various cancers (https://www.clinicaltrials.gov). Our study raises the possibility

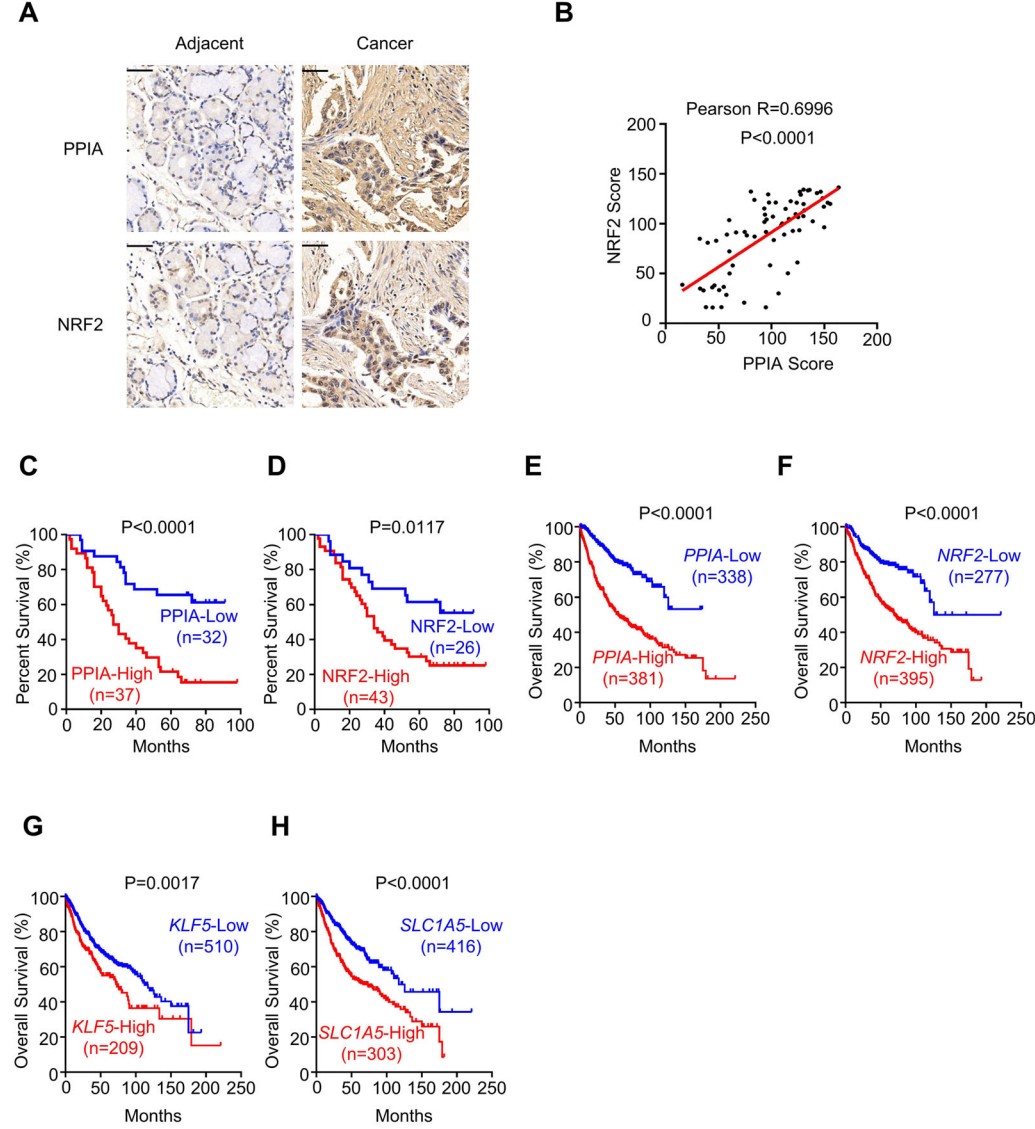

**Fig. 7 | Elevated PPIA and NRF2 accumulation is positively associated with poor prognosis in NSCLC patients. A** Representative immunohistochemical analysis of PPIA and NRF2 in human LUAD tissue microarray containing lung tumor tissues and adjacent normal lung tissues. Scale bar = 50 μm. **B** Scatter plots showing a positive correlation of PPIA and NRF2 expression in IHC analysis of human NSCLC cancer tissues (*n* = 69 samples). Linear regression with Pearson R and two-tailed P values are shown. **C** Kaplan-Meier survival curves of patients with NSCLC divided by high or low PPIA protein expression level according to IHC analysis (*n* = 69 samples).

**D** Kaplan-Meier survival curves of patients with NSCLC divided by high or low NRF2 protein expression level according to IHC analysis (*n* = 69 samples). **E–H** Kaplan–Meier survival curves of patients with NSCLC based on *PPIA*, *NRF2*, *KLF5* and *SLC1A5* gene expression level (for *PPIA*, *n* = 719 samples; for *NRF2*, *n* = 672 samples; for *KLF5*, *n* = 719 samples; for *SLC1A5*, *n* = 719 samples). Data are integrated from Kaplan–Meier plotter (http://kmplot.com/analysis/). Statistical significance for Kaplan–Meier survival curves (**C–H**) was calculated by Log-rank (Mantel-Cox) test. Source data are provided as a Source Data file.

that a deeper understanding of CsA's mechanism will inspire a more reasonable therapeutic regimen in future clinical investigations using new, optimized PPIA inhibitors to treat patients with NRF2-hyperactivated NSCLC.

## Table 3 | The correlation between PPIA and NRF2 protein expression in IHC analysis of human NSCLC cancer tissues

| The correlation between PPIA and NRF2 in human LUAD cancer tissues | | | |
|---|---|---|---|
| | PPIA | | |
| NRF2 | Low | High | P value |
| Low | 21 | 5 | P < 0.001 |
| High | 11 | 32 | |

The correlation between PPIA and NRF2 expression was analyzed by bivariate Pearson correlation analysis using SPSS Statistics 28.0 software (IBM, Armonk, New York). Statistically significant differences between two groups were determined by two-tailed test (P < 0.001).

During cancer progression, NRF2 signaling functions as a master mechanism in response to hostile metabolic microenvironment[55]. Several pioneering works have established that NRF2 activation in cancer cells leads to increased glutamine dependence via enhancing consumption of glutamate for fueling tricarboxylic acid cycle, glutathione synthesis and glutamate excretion[56–58]. To sustain intracellular glutamate content, NRF2-hyperactivated cells use glutamine transporter SLC1A5 to uptake greater extracellular glutamine, which is subsequently converted to glutamate by glutaminase[59]. The high requirement of glutamine in NRF2-hyperactivated cells indicates a potential dependency of NRF2-hyperactivated cells on SLC1A5[32,60]. SLC1A5 is up-regulated in the NRF2-hyperactivated cells, however, the underlying regulatory mechanisms of SLC1A5 expression by NRF2 are not fully understood. In this study, we demonstrated that oncogenic transcription factor KLF5, acting as a NRF2 target gene, activates the transcription of SLC1A5 via a specific GC-rich motif

sequence. While, given the pleiotropic roles of NRF2 activation, it will be intriguing to investigate whether other transcriptional or post-translational mechanisms contribute to NRF2-mediated SLC1A5 augmentation.

Recent studies have evidenced that KRAS-driven lung and pancreatic cancer with KEAP1 or NRF2 mutations are highly dependent on glutaminolysis and are vulnerable to glutaminase inhibition[32,61]. CB-839, a first-in-class glutaminase inhibitor, is actively explored as metabolic intervention for the treatment of multiple types of cancer, but its clinical benefit is limited as monotherapy[62]. Several phase I studies have demonstrated encouraging results for CB-839 as part of a combination regimen[63]. However, a recent phase II study combining CB-839 with pembrolizumab/chemotherapy as first-line treatment for KEAP1/NRF2-mutated metastatic NSCLC have been terminated due to lack of efficacy (NCT04265534)[64]. Despite these unsatisfactory outcomes, careful development and rational design of combination therapies with CB-839 might confer clinical benefits. Here, we have explored a combination therapy of CsA and CB-839. In both cell line-derived and patient-derived xenograft models with NRF2 hyper-activation, this combination treatment showed encouraging anti-cancer activity and tolerability. Our results may provide a rationale for exploiting CsA to sensitize patients with NRF2-hyperactivated tumors to glutaminase inhibition. However, extensive clinical investigations are required to optimize safe and effective combination therapy regimens of CsA and CB-839 to reduce drug-drug interaction and avoid drug resistance.

In summary, our study provides actionable mechanistic insights about NRF2 stability and reveals PPIA as a promising target for treating NRF2-hyperactivated lung cancer. Of note, the FDA-approved CsA, as a chemical inducer of NRF2 degradation, exhibits a significantly synergistic anti-cancer effect when administered as a combination therapy alongside the clinical glutaminase inhibitor CB-839 in NRF2-hyperactivated NSCLC. Our findings provide a potential framework toward precision medicine of this refractory cancer type in clinical trials.

## Methods

### Ethical statement
The experimental protocols of animal studies were approved by Institutional Animal Care and Use Committee (IACUC) of East China University of Science and Technology (ECUST-2021-10001).

### Cell cultures and reagents
HEK293T cells and lung cancer cell lines, including A549, H2030, H1299, H1395, H1650, H460, H1944, H1703, H1975, H2122, H23, H292, H358 and H838 were kindly provided by Stem Cell Bank, Chinese Academy of Sciences. EBC-1 was purchased from Shanghai Hongshun Biotechnology Co., Ltd. H441 and H522 cell lines were purchased from ATCC (Manassas, VA, USA). All the cell lines have been authenticated by STR (short tandem repeat) profiling and confirmed mycoplasma free by PCR-based assay. All cell lines were cultured in DMEM medium supplemented with 10% FBS, 100 unit/mL penicillin and 100 μg/mL streptomycin and kept in a 37 °C humidified incubator containing 5% $CO_2$.

The detailed information of reagents used in this study were provided in Supplementary Data 2. Antibody dilutions: NRF2 (Proteintech, 16396-1-AP; 1:1000 for WB; 2 μg for IP; 2 μg for ChIP; 1:100 for IHC), Histone-H3 (Proteintech, 17168-1-AP; 1:1000 for WB), PPIA (Proteintech, 10720-1-AP; 1:1000 for WB; 1:100 for IHC), Ubiquitin (Santa Cruz, sc-8017; 1:1000 for WB), SLC1A5 (Proteintech, 20350-1-AP; 1:1000 for WB), KLF5 (Proteintech, 21017-1-AP; 1:1000 for WB; 2 μg for ChIP), KEAP1 (Proteintech, 10503-2-AP; 1:1000 for WB), CRK (Proteintech, 16685-1-AP; 1:1000 for WB), TDP-43 (Proteintech, 10782-2-AP; 1:1000 for WB), HA (Proteintech, 51064-2-AP; 1:1000 for WB), Flag (Sigma-Aldrich, F3165; 1:1000 for WB; 2 μg for IP), GAPDH (Proteintech, 60004-1-Ig; 1:3000 for WB), IgG heavy chain (Proteintech, 16402-1-AP; 2 μg for IP).

### RIG-NRF2 Screen
To assess NRF2 protein degradation, we developed a fluorescence probe named RIG-NRF2 (DsRed-IRES-EGFP-NRF2). Briefly, NRF2 was fused with green fluorescent protein EGFP and co-expressed with red fluorescent protein DsRed within a single transcription unit, which was separated by an internal ribosome entry site (IRES) motif. The cDNA of human *NRF2* was amplified through polymerase chain reaction (PCR) using followed cloning primers:

FW: 5'-GATAATATGGCCACAATCGATATGATGGACTTGGAG-3'
RV: 5'-CACCATAGCGGCCGCCTCGAGGTTTTTCTTAACATC-3'.

Subsequently, the PCR product was cloned into the bicistronic lentiviral vector pLenti-DsRed_IRES_MAPT:EGFP vector (Addgene, Plasmid #92196) using one step cloning method.

HEK293T cells were transfected with RIG-NRF2 as well as packaging plasmids psPAX2 (Addgene plasmid #12260) and pMD2.G (Addgene plasmid #12259) using Lipofectamine 2000 reagent to generate lentivirus particles. A549 cells were then infected with lenti-virus and selected with G418. The generated A549 subclone stably expressing RIG-NRF2 was named as A549-RIG-NRF2. To detect NRF2 protein level, A549-RIG-NRF2 cells were seeded into 6-well plates at the confluence of 70%-80% and then treated with MG132, cycloheximide (CHX), Ki696, ML334 or compounds for indicated times. Then, treated cells were washed with PBS for 3 times and resuspended in PBS at the concentration of $1 \times 10^6$ cells/mL. Fluorescence signal was detected by BioTek neo2 multi-mode reader or flow cytometry (Beckman coulter Cytoflex LX).

### Plasmid construction
NRF2-K7R plasmid was kindly provided by Donna D. Zhang (University of Arizona, Tucson, USA) and the PCR-generated DNA containing NRF2-K7R was cloned into pCDNA3.1-Flag-Nanoluc using ClonExpress II One Step Cloning Kit. Site-direct mutagenesis and fragment deletion of *PPIA* and *NRF2* were performed using KOD-Plus-Mutagenesis Kit according to the manufacturer's instructions. Promoter region of *SLC1A5* (-2,000 to +100 bp) and promoter region of *KLF5* (-4,000 to 0 bp) were cloned into KpnI and XhoI sites of pGL3-Basic using ClonExpress II One Step Cloning Kit.

### Luciferase reporter assay
Antioxidant response element (ARE)-driven reporter gene expression was performed as described previously[65]. Briefly, cells were seeded into white 96-well plates (15,000 cells per well) and subsequently co-transfected with pGL4.37[luc2P/ARE/Hygro], pCDNA3.1-NRF2-Flag and pSV40 vector with a ratio of 2:1:0.5 (totally 100 ng DNA/well) using lipofectamine 2000 reagent (Invitrogen, Grand Island, NY, USA). After 4 h transfection, cells were treated with different concentrations of CsA for another 48 h. Luciferase activity was measured using a Dual-Glo® Luciferase Assay System (Promega, Madison, WI, USA) according to the manufacturer's instructions.

### Promoter activity assay
HEK293T cells were co-transfected with *KLF5* promoter luciferase construct and pCDNA3.1-Flag-NRF2 or *SLC1A5* promoter luciferase construct and pCDNA3.1-Flag-KLF5 using lipofectamine 2000. The transfected cells were seeded into 96-well plates and allowed to grow for 36 h. Subsequently, cells were lysed and the luciferase activities were detected by Dual-Luciferase Reporter Assay System (Promega, Madison, WI, USA) using BioTek neo2 multi-mode reader (BioTek, Winooski, VT, USA).

### CRISPR/Cas9 mediated *PPIA* gene knockout (*PPIA* KO)
*PPIA* gene knockout cancer cells were generated by using CRISPR/Cas9-mediated genome editing[66]. The lentiCRISPRv2 (Addgene, #52961) containing the specific sgRNA targeting *PPIA* gene and the packaging plasmid psPAX2 (Addgene plasmid #12260) and pMD2.G

(Addgene plasmid #12259) was co-transfected into HEK293T cells. Two distinct sgRNAs targeting human *PPIA* gene were designed as following: sg*PPIA*#1: TTCTTCGACATTGCCGTCGA, and sg*PPIA*#2: ACAAGGTCCCAAAGACAGC. Lentivirus infection of NSCLC cells was conducted in the presence of polybrene (8 μg/mL) for 12 h. *PPIA*-KO cells were selected using puromycin (5 μg/mL) after expansion for 72 h and re-plating at high dilution in the presence of 5 μg/mL of puromycin in order to obtain individual colonies. Ablation of *PPIA* was verified using immunoblot.

### Immunoblot analysis
Cells were washed with cold PBS twice and lysed in lysis buffer (supplemented with 1% protease inhibitor cocktail) at 4 °C. Protein from lysed cells was resolved by SDS-PAGE and transferred to PVDF membranes. After being blocked in 5% bovine serum albumin in TBST (120 mM Tris, 150 mM NaCl, 0.05% Tween20, pH=7.4) at room temperature for 1 h, the membranes were incubated with different primary antibodies overnight at 4 °C. Subsequently, the membranes were washed with TBST for 3 times and incubated with HRP-conjugated secondary antibodies at room temperature for 3 h. After washing with TBST for 4 times, the protein bands were detected by ECL Western Blotting Substrate Kit (Thermo Scientific, MA, US).

### Ubiquitination assay
Cells were pre-treated with MG-132 (10 μM) for 4 h and lysed in lysis buffer (50 mM Tris, 150 mM NaCl, 1% NP-40, 0.25% sodium deoxycholate, 1 mM EDTA pH=7.4, supplemented with 1% protease inhibitor cocktail). Cell lysates were subjected to immunoprecipitation using primary anti-NRF2 antibody at 4 °C overnight. The immunoprecipitated proteins were eluted by boiling with SDS-loading buffer at 100 °C for 5 min and detected by immunoblot analysis using specific antibody against Ubiquitin.

### NanoBRET Assay[67]
HEK293T cells were seeded into 96-well clear-bottom white plates (20,000 cells per well) and then co-transfected with pCDNA3.1-NRF2-Nanoluc and HaloTag-Ubiqiuitin with a ratio of 1:100 (totally 300 ng DNA/well) using lipofectamine 2000 reagent (Invitrogen, Grand Island, NY, USA). 24-h after transfection, cells were incubated with 100 nM HaloTag NanoBRET 618 Ligand (Promega, Madison, WI, USA) for 24 h at 37 °C. BRET signals were detected by adding Nanoluc substrate furimazine (10 μM) with BioTek neo2 multi-mode reader (BioTek, Winooski, VT, USA).

### Quantitative PCR (Q-PCR)
Total RNA of cells or tumor tissues was extracted with GenElute™ Total RNA purification Kit and cDNA was prepared using Hifair® II 1st Strand cDNA Synthesis SuperMix (Yeasen, Shanghai, China). Quantitative reverse-transcription PCR was conducted using Hifair® III One Step RT-qPCR SYBR Green Kit (Yeasen, Shanghai, China) according to its manufacturer's instructions. The fold change of genes was calculated by $2^{-\Delta\Delta CT}$ method taking GAPDH as housekeeping gene ($\Delta Ct = \Delta Ct^{target} - \Delta Ct^{GAPDH}$; $\Delta\Delta Ct = \Delta Ct^{sample} - \Delta Ct^{control}$) and normalized with control groups which were defined as 1.0. All primers sequences were listed in Supplementary Data 1.

### Cell viability assay
Cell viability was determined by using MTT (3-(4,5-dimethylthiazol-2-yl)-2,5-diphenyltetrazolium bromide) method as described previously[68]. In brief, cells were seeded into 96-well plates and then treated with indicated concentrations of compounds for 96 h. Subsequently, cells were incubated with MTT reagent with a final concentration of 0.5 mg/mL for 4 h at 37 °C. The generated formazan crystals were dissolved in 150 μL dimethyl sulfoxide (DMSO) and the optical density at 570 nm was examined using BioTek neo2 multi-mode reader (BioTek, Winooski, VT, USA). The combination index (CI) of drug combinations was calculated using CalcuSyn software as described previously[68]. CI values < 1, =1, and >1 indicate synergism, additive, and antagonism, respectively.

### 3D colony formation assay
For 3D colony formation assay[69], 0.7% and 1.2% Bacto™ agar solution (BD Biosciences, San Jose, CA) were prepared, respectively. The mixture of 2x cell culture medium and 1.2% agar (in a 1:1 ratio) was seeded into 6-well plates as the bottom layer and the mixture of 2x cell culture medium containing NSCLC cancer cells stably expressing mCherry and 0.7% agar (in a 1:1 ratio) was plated as upper layer. Fresh culture medium containing various concentrations of compounds was maintained over the upper layer and changed every 3 days. After growing for 2 weeks, colonies were imaged using the fluorescence microscope (Olympus, Japan).

### Co-immunoprecipitation (Co-IP) assay
HEK293T cells were transiently transfected with different expression vectors. After 36 h, cells were washed with cold PBS and lysed with lysis buffer (50 mM Tris pH 8.0, 150 mM NaCl, 10% glycerol, 1 mM EDTA, 50 mM NaF, and 0.1% NP-40) supplemented with 1% protease inhibitor cocktail. Endogenous proteins derived from NSCLC cells were also used for co-immunoprecipitation. The whole cell lysates were incubated with specific antibody and protein A/G Sepharose beads overnight at 4 °C. Subsequently, the immunocomplexes were washed with lysis buffer for 3 times at 4 °C and eluted with loading buffer at 100 °C for 5 min. Protein sample was resolved by SDS-PAGE and immunoblot.

### Oxygen consumption rate (OCR) analysis
The OCR was analyzed using Seahorse XFe96 Analyzer (Agilent Technologies) according to the manufacturer's instruction[70]. A total of $1.5 \times 10^4$ cells were seeded per well into 96-well cell culture plates (Agilent Technologies) and cultured with 10% FBS supplemented DMEM overnight at 37 °C incubator containing 5% $CO_2$. Next day, the medium was removed and replaced with XF DMEM Base Medium, pH=7.4 (Agilent Technologies) supplemented with 2 mM glutamine (Agilent Technologies) and the cells were incubated in 37 °C non-$CO_2$ incubator to balance the $CO_2$ level. Using the XFe96 Analyzer, OCR was measured in baseline conditions (basal OCR) and metabolic stress conditions (maximal OCR) induced by several metabolic drugs, including oligomycin (1 μM), FCCP (0.5 μM), and Rotenone/Antimycin (0.5/0.5 μM) using Seahorse XF Cell Mito Stress Test kit (Agilent Technologies). Each measurement consisted of 3 cycles: 3 min Mix, 0 min Wait and 3 min Measure. The OCR values were normalized according to cell number and calculated with WAVE software (Agilent Technologies).

### Chromatin immunoprecipitation (ChIP)
ChIP assay was performed using ChIP Assay Kit (Beyotime Biotechnology, Shanghai, China) according to the manufacturer's protocol[71]. $5 \times 10^6$ cells were harvested and fixed with 1% formaldehyde for 10 min at 37 °C. The reaction was quenched by 125 mM glycine for 10 min at room temperature. The DNA of cell nuclei was isolated and sheared to 200-500 bp through sonication. Fragmented chromatin was subjected to immunoprecipitation reactions using Protein A/G agarose beads and anti-NRF2 (5 μg), anti-KLF5 (5 μg) or control IgG (5 μg) at 4 °C. The beads were then washed with Low-Salt Immune Complex Wash Buffer, High-Salt Immune Complex Wash Buffer, LiCl Immune Complex Wash Buffer and TE Buffer (twice) for 3 min at 4 °C. DNA-protein complexes were de-crosslinked by treating with 0.2 M NaCl and heating (65 °C, 4 hours). The proteins were removed by protein K treatment. The resultant DNA product was subjected to PCR analysis. The human *KLF5* or *SLC1A5* promoter-specific primers sequences were shown in Supplementary Data 1.

## siRNA transfection

siRNA negative control and siRNA targeting *NRF2, PPIA, KLF5* were obtained from Shanghai Synbio Technologies. For siRNA transfection, cells were seeded at about 60% confluence into 6 cm dishes and then transfected with specific siRNA using lipofectamine 3000 reagent following the manufacturer's manual for 36 h. The gene knockdown efficiency was confirmed by immunoblot. The siRNA sequences were listed in Supplementary Data 1.

## Recombinant protein expression and purification

Recombinant PPIA protein was expressed in BL21 (DE3) E. coli after induction with 0.2 mM IPTG (isopropyl-β-D-thiogalactopyranoside) overnight at 25 °C[72]. $^{15}$N-labeled PPIA protein was produced from E. coli grown at 16 °C. Cell pellets were collected and lysed by sonication in lysis buffer (25 mM Tris, 300 mM NaCl, 5% glycerol, pH=8.0). Then, the clarified lysate was loaded onto the Ni-NTA column (Thermofisher, MA, US) and washed with lysis buffer containing 20 mM imidazole. The bound target protein was eluted with lysis buffer containing 200 mM imidazole.

## Pull-down assay

Purified PPIA protein was conjugated with CNBr-activated Sepharose 4B beads (GE Healthcare, UK) according to standard manufacturer's protocol[72]. Firstly, the CNBr-activated Sepharose 4B beads were suspended and washed in 1 mM HCl and then washed with coupling buffer, 0.1 M NaHCO$_3$ pH=8.3 containing 0.5 M NaCl. Secondly, activated beads were incubated with or without purified PPIA protein in coupling buffer overnight at 4 °C and subsequently washed with coupling buffer and blocked with 0.1 M Tris pH=8.0 for 2 h. Thirdly, protein-conjugated beads were washed five cycles of different pH washing buffers (buffer 1, 0.1 M acetate, 0.5 M NaCl, pH 4.0; buffer 2, 0.1 M Tris, 0.5 M NaCl, pH 8.0) and incubated with cell lysate in binding buffer (25 mM HEPES, 0.15 M NaCl, 1 mM DTT, pH=7.5) for 3 h at 4 °C. Lastly, the beads were washed with binding buffer for 5 times and eluted with SDS gel-loading buffer for 5 min at 100 °C followed by immunoblot analysis.

## Glutamine assay

Treated cells were plated at 20,000 cells/well in 96-well plates in DMEM medium supplemented with 4 mM glutamine and 10% dialyzed FBS. Cellular glutamine level was measured using the glutamine/glutamate-Glo Assay kit (Promega, Madison, WI, USA) following manufacturer's instruction.

## Stable-isotope labeling and metabolic flux analysis[73]

For $^{13}$C$_5$-glutamine labeling, treated cells (2 × 10$^6$) were cultured in glutamine free DMEM medium containing 10% dialyzed serum and 2 mM $^{13}$C$_5$-glutamine for 8 h. After labeling, cells were washed twice with 37 °C pre-warmed saline, lysed by 1 mL of 80% ice-cold methanol solution, and sonicated five times for 5 s, with 5 s intervals between treatments. The samples were centrifuged at 18,000 g for 10 min at 4 °C and all the supernatant were collected and stored at -80 °C until quantitative analysis.

Metabolites were quantitated by Metabo Profile Biotechnology (Shanghai, China). Samples were thawed on an ice bath to reduce sample degradation before processing. 400 μL 80% methanol solution was added to cell samples. Then, the samples were sonicated and centrifuged at 18,000 *g* for 15 minutes at 4 °C. All the supernatant after centrifugation was collected for LC-MS analysis.

For LC-MS analysis, the samples were analyzed by ultra-performance liquid chromatography coupled to tandem mass spectrometry (UPLC-MS/MS) system (ACQUITY UPLC-Xevo TQ-S, Waters Corp., Milford, MA, USA). The optimized instrument settings are briefly described as follows. For HPLC, column: UPLC HILIC (4.6-mm column, Amide XBridge, Waters), column temperature: 40 °C, sample

manager temperature: 10 °C, mobile phases: A = 5% acetonitrile in water (with 20 mM ammonium hydroxide and ammonium acetate); and B = acetonitrile, gradient conditions: 0-3.5 min (85-32% B), 3.5-12 min (32-2% B), 12-16.5 min (2% B), 16.5-17 min (2-85% B), 17-25.5 min (85% B), flow rate: 0.40 mL/min, and injection vol.: 5.0 μL. For mass spectrometer, capillary 2.5 (ESI-) Kv, source temperature: 150 °C, desolvation temperature: 550 °C, and desolvation gas flow: 1000 L/h. The raw data files generated by UPLC-MS/MS were processed using the MassLynx software (v4.1, Waters, Milford, MA, USA) to perform peak integration, calibration, and quantitation for each metabolite.

## Nuclear magnetic resonance spectroscopy

[$^1$H, $^{15}$N] HSQC (heteronuclear single quantum coherence) experiments of $^{15}$N-labeled PPIA were performed in the presence or absence of 5-fold molar excess of NRF2 fragment PBM $^{169}$VAQVAPVD$^{176}$ at 25 °C on a Bruker Avance III 600 MHz NMR spectrometer equipped with a cryogenically cooled probe. All of the NMR samples were prepared using buffer containing 100 mM NaCl, 50 mM Bis-Tris, pH 7.4 with 10% D$_2$O, and the final concentration of PPIA was 500 μM. NMR data processing and analysis were performed by using the programs of NMRPipe1 and Sparky (Goddard and Kneller, Sparky 3, University of California, San Francisco). The amide resonance assignments of PPIA extracted from the BMRB entry of 27620 were used in the chemical shift perturbation (CSP) analysis, and the CSP values (Δδ) for $^{15}$N and $^1$H nuclei were calculated according to Eq. 1, where $\Delta\delta_N$ and $\Delta\delta_H$ represent the CSP values of the amide nitrogen and proton, respectively.

$$\Delta\delta = \sqrt{\left((\Delta\delta_N/5)^2 + \Delta\delta_H{}^2\right)/2} \tag{1}$$

## Crystallization and structure determination

For crystallization, the recombinant PPIA protein was further purified through gel filtration (Superdex 75, GE Healthcare) in buffer containing 25 mM Hepes, 100 mM NaCl, 0.5 mM TCEP with pH value of 7.5. The purified protein was concentrated into 50 mg/mL and incubated with indicated NRF2 fragment PBM ($^{169}$VAQVAPVD$^{176}$, 2 mM) at 4 °C for 2 h. Hanging-drop vapor-diffusion in 24-well crystallization plates (Hampton Research, USA) was used for co-crystallization. The PPIA/NRF2-PBM complex was mixed with equal volume of reservoir solution consisting of 25 mM Hepes, 25% PEG 3350, 5 mM TCEP with pH value of 7.5. The cubic complex crystal appeared within two weeks and then mounted in a 0.3 μm loop with addition cryoprotectant of 30% Tween 20 and flash-cooled in liquid nitrogen before data collection. X-ray diffraction data was collected at the synchrotron beamline BL19U1 in SSRF, Shanghai, China. The data was processed using HKL3000 program suite (HKL Research, Charlottesville, VA), phased by Molrep (CCP4) with PDB entry 3K0M as search model and further refined via Phenix as well as Coot. Data quality and processing statistics were summarized in Table 2 and Supplementary Table 2. The images of complex structure were prepared using PyMOL (Molecular Graphics System, Version 2.2.3; Schrödinger, LLC). The structure factors of the final complex crystal were deposited in the Protein Data Bank (PDB, www.rcsb.org, PDB ID: 8HZ8).

## Mouse strains

Female BALB/c Nude mice were purchased from Shanghai Jiesijie Laboratory. Female NOD/ShiLtJGpt-*Prkdc*$^{em26Cd52}$*Il2rg*$^{em26Cd22}$/Gpt (NCG) mice and female CD34$^+$ HSC-derived humanized NCG mice were purchased from Jiangsu GemPharmatech Company. Mice were housed in a dedicated animal facility designed for laboratory rodents. The temperature in the facility was maintained at a range of 20-24 °C with a relative humidity of 40-60%. A 12-hour light-dark cycle was maintained. All maintenance and experimental procedures involving animals were

approved by the Institutional Animal Care and Use Committee of East China University of Science and Technology. The maximal tumor size permitted by the Institutional Animal Care and Use Committee was 3000 mm³.

## Xenograft models in nude mice
A549 and H1650 cells with inducible sh*Ctrl* or sh*PPIA* were injected subcutaneously at the back of 6-week-old female BALB/c nude mice (5×10⁶ per mouse). The mice were given doxycycline (20 mg/kg) by oral gavage every other day when tumor volumes reached approximately 50 mm³. Tumor volumes were measured with calipers every 3 days till the control group volumes reached to around 500 mm³.

## Xenograft models in humanized NCG mice
A549 cells (5×10⁶ per mouse) were subcutaneously injected at the back of 18-week-old female CD34⁺ HSC-derived humanized NCG mice. When tumor volume reached to around 50 mm³, the mice were randomly divided into four groups: (1) Vehicle; (2) CB-839 (CB-839 in 25% (W/V) β-cyclodextrin pH=2, 150 mg/kg, p.o. twice daily); (3) CsA, (CsA in 10% DMSO + 10% Kolliphor EL + 80% PBS, 20 mg/kg i.p. every 3 days); (4) CB-839/CsA combination (CB-839 150 mg/kg, p.o. twice daily; CsA 20 mg/kg, i.p. every 3 days). Tumor volumes were measured with calipers every 3 days till the control group volumes reached to around 500 mm³.

## Lung orthotopic model
Lung orthotopic model was established using luciferase-expressing A549 (A549-Luc) cells[74]. In brief, 6-week-old female BALB/c nude mice were injected with 100 μL A549-Luc cells (5×10⁶ per mouse) through the lateral tail vein and lung lesion formation was monitored by bioluminescence imaging using D-luciferin (2 mg per mouse) every 7 days. Mice were randomly divided into four groups: (1) Vehicle; (2) CB-839 (CB-839 in 25% (W/V) β-cyclodextrin pH=2, 150 mg/kg, p.o. twice daily); (3) CsA, (CsA in 10% DMSO + 10% Kolliphor EL + 80% PBS, 20 mg/kg i.p. every 3 days); (4) CB-839/CsA combination (CB-839 150 mg/kg, p.o. twice daily; CsA 20 mg/kg, i.p. every 3 days). The body weight of each mouse was recorded weekly. At the end of the experiments, mice were euthanized and lung tissues were harvested, and fixed with bouin's fluid. The number of lung nodules was recorded.

## Patient-derived xenograft (PDX)
Human tumor specimens were supplied by Affiliated Tumor Hospital of Guangxi Medical University. Written informed consent was provided by participants and approved by Ethics Committee of Guangxi Medical University Cancer Hospital. The genomic DNA of tumor tissues derived from patients with lung adenocarcinoma was extracted using TIANamp Genomic DNA Kit (QIAGEN). Mutations in *KRAS* (exons 2 and 3) and *KEAP1* (exons 2, 3, 4, 5) were analyzed using PCR amplification and DNA sequencing. Primers sequences were listed in Supplementary Data 1.

PDX study was performed as described previously with some modifications[75]. Briefly, tumor fragments were implanted at the back of 6-week-old female NCG mice. When tumor volumes reached approximately 1,000 mm³, tumor tissues were collected and reimplanted into NCG mice. When the third generation of tumors reached 150-200 mm³, those mice were randomly divided into 4 groups: (1) Vehicle (p.o. twice daily); (2) CB-839 (CB-839 in 25% (W/V) β-cyclodextrin pH=2, 150 mg/kg, p.o. twice daily); (3) CsA (CsA in 10% DMSO + 10% Kolliphor EL + 80% PBS, 20 mg/kg i.p. every 3 days); (4) CB-839/CsA combination (CB-839 150 mg/kg, p.o. twice daily; CsA 20 mg/kg, i.p. every 3 days). Mice were sacrificed when the vehicle group volumes reached to around 2000 mm³. Tumor volumes were measured with calipers every 3 days.

## Tumor tissue microarray
Tissue microarray of human LUAD samples (n = 69) were obtained from Shanghai Superbiotek Pharmaceutical Technology Co. Ltd. (Shanghai, China). For immunohistochemical staining, the tissue microarray was dewaxed and rehydrated. Protein expression was detected by anti-PPIA or anti-NRF2 antibodies at 4 °C overnight and followed by HRP-labeled secondary antibodies. Subsequently, tissue microarray was probed by streptavidin-alkaline phosphatase system and scanned using 3DHISTECH (Hungary, PANNORAMIC Desk/MIDI/250/1000).

## Bioinformatics analysis
The association of the gene expression level of *KLF5* with that of *SLC1A5* or *NRF2* in NSCLC patients was analyzed in PrognoScan database (dataset GSE8894, http://www.prognoscan.org)[27,76]. To further examine the clinical significance of *PPIA*, *NRF2*, *KLF5*, and *SLC1A5* expression, we assessed whether their expression levels were associated to survival benefits in NSCLC using Kaplan-Meier survival plot for grouping patients (Kaplan-Meier plotter, http://kmplot.com/analysis/)[36]. The risk differences were calculated by the log-rank test.

## Statistics and reproducibility
Statistical analysis was conducted using GraphPad Prism 8.0 software (GraphPad Software, California, USA). Specially in Table 3, the correlation between NRF2 and PPIA expression was analyzed by bivariate Pearson correlation analysis using SPSS Statistics 28.0 software (IBM, Armonk, New York). Data were presented as mean ± S.D. for the specified number of independent experiments. Statistical significance between two groups were determined by unpaired two-tailed Student's t-test. For Kaplan-Meier survival curves, Log-rank (Mantel-Cox) test was performed. For tumor growth inhibition, One-way ANOVA analysis was performed. The Pearson's correlation method was utilized to perform the correlation test. $P < 0.05$ indicates statistically significant, $P < 0.01$ indicates very significant, $P < 0.001$ indicates highly significant. No data were excluded from the analyses.

## Reporting summary
Further information on research design is available in the Nature Portfolio Reporting Summary linked to this article.

## Data availability
The atomic coordinates and experimental data have been deposited in the Protein Data Bank (www.wwpdb.org) under accession code 8HZ8. The PPIA-CsA X-ray crystallographic data and PPIA-PBM search model used in this study are available in the Protein Data Bank (www.wwpdb.org) under accession code 1CWA and 3K0M. The publicly available NSCLC clinical data used in this study are available in the GEO database under accession code GSE8894[27]. The remaining data associated with this study are present within the Article, Supplementary Information or Source Data file. Source data are provided with this paper.

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

## Acknowledgements

This project was supported by the National Natural Science Foundation of China (81773775, H.J., 82173857, H.J., 82373913 H.J.) and Shanghai Committee of Science and Technology (22S11900900, H.J.), Shanghai Frontiers Science Center of Optogenetic Techniques for Cell Metabolism and State Key Laboratory of Drug Research, SKLDR-2023-KF-04. We thank Prof. Kangdong Liu (Zhengzhou University) for technical assistance of PDX study. We thank Prof. Naixia Zhang (Shanghai Institute of Material Medica) for technical assistance of NMR study. We thank Prof. Hua Lu (Tulane University) for critical reading and editing of our manuscript. We thank Dr. Wenming Qin and Dr. Hai Wu at SSRF beamline BL19U for assistance with data collection. We thank the staff members of the Large-scale Protein Preparation System at the National Facility for Protein Science in Shanghai (NFPS), Zhangjiang Lab, Shanghai Advanced Research Institute, Chinese Academy of Science, China for providing technical support and assistance in data collection and analysis.

## Author contributions

This work was conceptualized by W.L. and J.H. PDX models and lung orthotopic models were performed by J.C., W.W., Q.H., T.G., Q.W., Y.Xue., X.Y. and H.J. Cell culture, qPCR, confocal immunoblot analysis, ChIP assay were done by J.C., Q.H., T.G., and Y.L. Immunoblot analysis, luciferase reporter, pull down and Co-IP experiments were performed by J.C., Q.H., X.L., and T.G. Plasmid construction, protein expression and purification were performed by T.G., and Y.L. X-ray co-crystal structure, NMR assay was performed by W.W., H.M., N.Z., Y. Xu., and W.L. Bioinformatics analysis were done by J.C., and Q.H. Data analysis and interpretation were done by W.L., J.C., W.W., Q.H., T.G., Y.L., H.M., B.F., Q.W., M.L., R.N., F.C., and J.H. The manuscript was drafted by W.L., J.C., W.W., Q.H., and J.H. and reviewed and modified by all the authors.

## Competing interests

The authors declare no competing interests.

## Additional information

[1]Shanghai Frontiers Science Center of Optogenetic Techniques for Cell Metabolism, Shanghai Key Laboratory of New Drug Design, School of Pharmacy, East China University of Science and Technology, Shanghai, China. [2]Shanghai Key Laboratory of Regulatory Biology, Institute of Biomedical Sciences and School of Life Sciences, East China Normal University, Shanghai, China. [3]State Key Laboratory of Cell Biology, Shanghai Institute of Biochemistry and Cell Biology, Center for Excellence in Molecular Cell Science, Chinese Academy of Sciences, Shanghai, China. [4]School of Life Science, Hangzhou Institute for Advanced Study, University of Chinese Academy of Sciences, Hangzhou, China. [5]Department of General Surgery, Ruijin Hospital, Shanghai Jiao Tong University School of Medicine, Shanghai, China. [6]Key Laboratory of Early Prevention and Treatment for Regional High Frequency Tumor, Ministry of Education, Nanning, China. [7]Guangxi Medical University Cancer Hospital, Nanning, China. [8]Shanghai Institute of Materia Medica, Chinese Academy of Sciences, Shanghai, China. [9]Computational Structural Biology Section, Basic Science Program, Frederick National Laboratory for Cancer Research, National Cancer Institute at Frederick, Frederick, USA. [10]Department of Human Molecular Genetics and Biochemistry, Sackler School of Medicine, Tel Aviv University, Tel Aviv, Israel. [11]Genomic Medicine Institute, Lerner Research Institute, Cleveland Clinic, Cleveland, USA. [12]These authors contributed equally: Weiqiang Lu, Jiayan Cui, Wanyan Wang, Qian Hu, Yun Xue. ✉e-mail: wqlu@bio.ecnu.edu.cn; huangjin@ecust.edu.cn

