## [Peer Review File · Nature Communications]

Reviewers' Comments:

Reviewer #1:

Remarks to the Author:

The authors have done an excellent job addressing my concerns and the manuscript is now suitable for publication.

Reviewer #2:

Remarks to the Author:

The authors have addressed the majority of the prior concerns. However, they have not tested the combination of CsA +/-CB839 in any immunocompetent models which is important given the known immunosuppressive roles of CsA. Any effects seen in these immunodeficient models may be completely different in immunocompetent animals.

- The authors need to test the efficacy of CsA +/-CB839 in immune-competent transplant or autochthonous lung cancer models with KEAP1 or NRF2 mutations.

- The study overall lacks in vivo rigor in validating the NRF2/PPIA/CsA/glutamine-axis. The in vivo experiment now provided in figure S4J does not provide insight into the importance of PPIA in the growth of NRF2 high tumors because: 1) there is no control cell line with WT KEAP1; and 2) PPIA is required for the growth of A549 cells in vitro based on Figure 4I, therefore cell viability is compromised prior to transplantation which does not really allow authors to test actual tumor growth. The experiment needs to be repeated with inducible shRNA or sgRNA against PPIA. Furthermore, the CsA is truly acting through the disruption of NRF2/PPIA then treatment with CsA in vitro AND in vivo should not have any additional effect in cells with PPIA knockdown or knockout.

Reviewer #3:

Remarks to the Author:

In their manuscript "Inhibition of Isomerase PPIA Impairs NRF2 Protein Stability and Undermines NRF2-Driven Glutamine Addition in Non-Small Cell Lung Cancer" the authors attempt to demonstrate that a synergistic response of PP1A(cyclophilin A) and the glutamine transporter SLC1A5 can reduce tumor growth. With extensive and exhaustive efforts to address many of the reviewers' technical questions from the first round of review, I highlight the areas that still remain a gap in understanding and other points that could address the impact of this work in the clinic.

The authors argue that abnormal Nrf2 activation can be a pivotal oncogenic driver of lung cancer. The purpose of this work is to identify an upstream target of the Nrf2 pathway that can be targeted to enhance Nrf2 degradation, thus reduce its oncogenic effect. Nrf2 is a transcription factor, direct activation of which is not readily tractable with small molecule therapies.

Certain NSCLC cancer cell lines are driven by mutations in KRAS and KEAP1 and the authors identify both PP1A and the glutamine transporter SLC1A5 as being key targets for these cell lines. The authors argue that KRAS mutants potentiate the transcriptional activation of Nrf2, thus work with Keap1 mutations, which further activate Nrf2.

The author's end goal is showing that by co-targeting both PP1A as well as a glutaminase, they can inhibit the tumor growth of NSCLC in a Xenograft model (inject A549-luc cells). Thus the

potentiation effect of Nrf2 activation is important, but only in combination with inhibition of glutamine metabolism.

The authors have addressed many of the reviewer1's technical comments adequately, but Reviewer 1 implies that there may also be a secondary effect of inhibiting PP1A (as an essential gene) to initiate Nrf2 degradation, rather than a direct role in binding. The authors do not adequately address the concern that this is part an indirect effect. The established mechanism of action of CsA is to bind to cyclophilin A (PPIA) and restrict dephosphorylation of the NFAT family of genes (not the ARE-responsive genes that are regulated by Nrf2).

Also the reviewer points out a gap in that inhibition of PP1A alone is insufficient to reduce tumor growth, and the authors must amplify this effect with the inhibition of glutaminase. The argument is that PP1A KO results in sensitization of cells to glutamine starvation, but that inhibition of PP1A alone or glutaminase inhibition alone is insufficient.

Reviewer 2 requested a more thorough glutamine metabolism tracing studies on cell viability to show that PPIA K/O can decrease the flux of glutamine-derived TCA metabolites, which the authors provide. Reviewer 2 also implies that CsA as an immunosuppressive may be an issue and that future in vivo studies should be performed in an immunocompetent mouse model (humanized immune system).

In general, I feel that this paper is interesting scientifically but there is not sufficient discussion of how this is really a biological study with tools but that these tools may not be translatable to the clinic.

First— There are still gaps in understanding of mechanism:

How does Nrf2's interaction with PP1A prevent its interaction with Keap1 and subsequent ubiquitination, given that the interaction site for Keap1 and for PP1A are not overlapping? The identified region 169VAQVAPVD176 between the Neh4 and Neh5 domains differs from the Neh2 binding site for Keap1 (76LDEETGEFL84) so the binding does not appear to be competitive with Keap1.

The authors demonstrate through domain truncation experiments to narrow down the potential binding site and then by mutation of P174A they can reduce Nrf2 levels. The cyclophilinA consensus binding motif is FGP X Lp. Would mutation of the Nrf2 sequence between the Neh4 and Neh5 consensus domains to the strongest Cyclophilin binding motif increase Nrf2 levels?

Second: Activation of the Nrf2 pathway can occur through adaptive response to electrophiles, such as BG12 (dimethylfumarate). Have the authors tested a combination of dimethyl fumarate with telaglenastat for glutamine metabolism inhibition on the growth of tumor cells in the Xenograft model?

Third: The established mechanism of action of CsA is to bind to cyclophilin A (PPIA) and restrict dephosphorylation of the NFAT family of genes (not the ARE-responsive genes that are regulated by Nrf2). Are there additional cross-talk mechanisms between the NFAT and ARE transcription factor cascades besides the direct binding of cyclophilin A to Nrf2? For example, there are several known links between the ARE-responsive pathway and the NF-κB responsive pathway. The 'N-factors' in pancreatic cancer: functional relevance of NF-κB, NFAT and Nrf2 in pancreatic cancer | Oncogenesis (nature.com)

Fourth: CsA itself is known to cause cancer in high doses (lymphoma and skin cancer), prompting the need for second generation inhibitors. Note FK506 or the more selective L-732531 is a structurally unrelated inhibitor of calcineurin, as is VIVIT (a peptide that interferes with the calcineurin-NFAT interaction). Lastly, there are less toxic analogues of cyclosporin A (ISATX247) The 'N-factors' in pancreatic cancer: functional relevance of NF-κB, NFAT and Nrf2 in pancreatic cancer - PMC (nih.gov).

Have the authors demonstrated that these other tools work in the same way as CsA on Nrf2 binding, as a potential therapy in the clinic without the cancer-causing attributes of CsA?

It would be interesting to see if the route of administration of these drugs to the lungs (breathing) might have a faster PD response and limit exposures that could cause other cancers.

Fifth: There are many clinical trials utilizing CB-839 (telaglenastat) along with antibodies for multiple tumor types. There is a disclosure at 2020 ASCO of Phase II trial of KEAPSAKE trial--telalenasat with pembrolizumab (Keytruda)(standard of care chemotherapy) for NSCLC which apparently terminated for lack of clinical benefit. This information could be included in the discussion to clarify the importance of the combination therapy vs CB-839 alone.

A phase II randomized study of telaglenastat, a glutaminase (GLS) inhibitor, versus placebo, in combination with pembrolizumab (Pembro) and chemotherapy as first-line treatment for KEAP1/NRF2-mutated non-squamous metastatic non-small cell lung cancer (mNSCLC). | Journal of Clinical Oncology (ascopubs.org)
Study Record | Beta ClinicalTrials.gov

Fifth: Combination therapy with two small molecules would require extensive clinical studies on drug-drug interactions, particularly to avoid tumor cell drug resistance.

In general, this paper appears to be scientifically grounded and should be published, but the overall context of the work being performed in the clinic and context of the Phase II studies that have failed would improve the translatability of this work.

Point-by-Point Response Letter

REF 1.1 – Excerpt from the reviewer’s comment or summary of this point – *the manuscript is now suitable for publication.*

Reviewer Comment

The authors have done an excellent job addressing my concerns and the manuscript is now suitable for publication.

Author Response

We are deeply appreciative of the Reviewer’s positive comment and recommendation for publication of our study.

REF 2.1 – Excerpt from the reviewer’s comment or summary of this point – The authors have addressed the majority of the prior concerns.

Reviewer Comment

The authors have addressed the majority of the prior concerns. However, they have not tested the combination of CsA +/-CB839 in any immunocompetent models which is important given the known immunosuppressive roles of CsA. Any effects seen in these immunodeficient models may be completely different in immunocompetent animals.

The authors need to test the efficacy of CsA +/-CB839 in immune-competent transplant or autochthonous lung cancer models with KEAP1 or NRF2 mutations.

Author Response

We thank the Reviewer for pointing out this issue. We have now examined the efficacy of the combination of CsA +/-CB839 using a humanized NCG mouse model implanted with human lung cancer cell line A549 (*KEAP1* mutant). Humanized NCG mice were established by engrafting with CD34⁺ hematopoietic stem cells (HSC) isolated from human cord blood. The reconstitution of human immune subpopulations was confirmed by flow cytometry analysis (Figure R1A). As shown in Figure R1B, the CsA and CB-839 monotherapies each impair tumor growth, with inhibition rates of $44.40 \pm 13.90\%$ and $51.10 \pm 7.85\%$, respectively, compared to the vehicle control. The CsA and CB-839 combination therapy performed superior, showing an inhibition rate of $66.28 \pm 8.11\%$. Of note, the efficacy of the combination of CsA +/-CB839 in immune-competent humanized NCG mice was comparable with that of patient-derived xenograft (PDX) or cell line-derived xenograft (CDX) models in immunodeficient mice in our previous manuscript (Figure 6F). Together, these findings demonstrate that the combination of CsA +/-CB839 retained anti-tumor efficacy in immune-competent mice.

To be clear, we have added these new experimental results (Figure S11B) and detailed explanations in the revised manuscript.

A**B**
Figure R1. Drug combination of CsA and CB-839 can inhibit growth of A549 xenograft tumors in CD34⁺ HSC-derived humanized mice. (A) Representative flow cytometry analysis of mouse CD45⁺ cells and human CD45⁺ cells out of leukocytes (left) and human CD3⁺ cells and CD19⁺ cells out of human CD45⁺ cells (right) in CD34⁺ HSC-derived humanized mice at 12 weeks post-transplantation. (B) Tumor growth in CD34⁺ HSC-derived humanized mice bearing A549 xenograft tumors treated with the vehicle (p.o. twice daily), CB-839 (150 mg/kg, p.o. twice daily), CsA (20 mg/kg, i.p. every 3 days), and CB-839 and CsA drug combination.

REF 2.2

Reviewer Comment

- The study overall lacks in vivo rigor in validating the NRF2/PPIA/CsA/glutamine-axis. The in vivo experiment now provided in figure S4J does not provide insight into the importance of

PPIA in the growth of NRF2 high tumors because: 1) there is no control cell line with WT KEAP1; and 2) PPIA is required for the growth of A549 cells in vitro based on Figure 4I, therefore cell viability is compromised prior to transplantation which does not really allow authors to test actual tumor growth. The experiment needs to be repeated with inducible shRNA or sgRNA against PPIA. Furthermore, the CsA is truly acting through the disruption of NRF2/PPIA then treatment with CsA in vitro AND in vivo should not have any additional effect in cells with PPIA knockdown or knockout.

Author Response

To address the Reviewer's concern, we have now established a doxycycline-inducible sh*PPIA* expression system in A549 cells (*KEAP1* mutant) and H1650 cells (*KEAP1* WT) by using a Tet-On lentiviral vector (Clontech; Mountain View, CA, USA). We found that treatment with doxycycline effectively induced PPIA knockdown in both A549 and H1650 cells with inducible sh*PPIA* (Figure R2A). As expected, PPIA knockdown resulted in profound reduction in the NRF2 protein levels in A549 cells with inducible sh*PPIA* (Figure R2A), but not in H1650 cells (low basal NRF2 levels) with inducible sh*PPIA*.

We then employed A549 and H1650 cells with inducible sh*Ctrl* or sh*PPIA* to generate tumors in mice. Doxycycline (20 mg/kg) was administered to induce shRNA expression by oral gavage. Notably, doxycycline-induced PPIA knockdown significantly reduced growth of A549 xenograft tumors, but failed to affect growth of H1650 xenograft tumors (Figure R2B and R2E). Moreover, doxycycline-induced PPIA knockdown and CsA exhibited comparable anti-proliferation efficacy against A549 cells *in vitro* and *in vivo*, whereas their combination failed to further improve the proliferation suppression activity, suggesting that CsA acts through targeting PPIA (Figure R2C and R3D). Together, these results further demonstrate the importance of PPIA in the growth of NRF2 high tumors.

To be clear, we have added these new experimental results (Figure S4J) and detailed explanations in the revised manuscript.

A**B****C****D****E**
Figure R2. (A) Representative immunoblot analysis of PPIA and NRF2 in A549 and H1650 cells with inducible sh*Ctrl* or sh*PPIA* treated with doxycycline (1 $\mu\text{g}/\text{mL}$) for 48h. (B) A549 and H1650 with inducible sh*Ctrl* or sh*PPIA* were used to generate subcutaneous tumors in mice. When tumor volumes reached approximately 50 mm^3 , doxycycline (20 mg/kg) was orally administrated to mice every other day. Tumor volume was measured every three days using slide calipers (n = 6). (C) Relative cell growth of A549 with inducible sh*Ctrl* or sh*PPIA* upon CsA treatment (n = 3). Quantitative results were shown in upper panel and representative colony formation images were presented in lower panel. (D) A549 subcutaneous tumors with inducible sh*PPIA* were treated with vehicle or CsA (20 mg/kg , i.p. every 3 days). When tumor volumes reached approximately 50 mm^3 , doxycycline (20 mg/kg) was orally administrated to mice every other day. Tumor volumes were measured every three days using slide calipers (n = 6). (E) Representative immunoblot analysis of PPIA and NRF2 in A549 subcutaneous tumors from Figure R2B and R2D.

REF 3.1 – Excerpt from the reviewer’s comment or summary of this point – In general, this paper appears to be scientifically grounded and should be published

Reviewer Comment

In their manuscript “Inhibition of Isomerase PPIA Impairs NRF2 Protein Stability and Undermines NRF2-Driven Glutamine Addition in Non-Small Cell Lung Cancer” the authors attempt to demonstrate that a synergistic response of PPIA (cyclophilin A) and the glutamine transporter SLC1A5 can reduce tumor growth. With extensive and exhaustive efforts to address many of the reviewers’ technical questions from the first round of review, I highlight the areas that still remain a gap in understanding and other points that could address the impact of this work in the clinic.

The authors argue that abnormal Nrf2 activation can be a pivotal oncogenic driver of lung cancer. The purpose of this work is to identify an upstream target of the Nrf2 pathway that can be targeted to enhance Nrf2 degradation, thus reduce its oncogenic effect. Nrf2 is a transcription factor, direct activation of which is not readily tractable with small molecule therapies.

Certain NSCLC cancer cell lines are driven by mutations in KRAS and KEAP1 and the authors identify both PPIA and the glutamine transporter SLC1A5 as being key targets for these cell lines. The authors argue that KRAS mutants potentiate the transcriptional activation of Nrf2, thus work with Keap1 mutations, which further activate Nrf2.

The author’s end goal is showing that by co-targeting both PPIA as well as a glutaminase, they can inhibit the tumor growth of NSCLC in a Xenograft model (inject A549-luc cells). Thus the potentiation effect of Nrf2 activation is important, but only in combination with inhibition of glutamine metabolism.

The authors have addressed many of the reviewer1’s technical comments adequately, but Reviewer 1 implies that there may also be a secondary effect of inhibiting PPIA (as an essential gene) to initiate Nrf2 degradation, rather than a direct role in binding. The authors do not adequately address the concern that this is part an indirect effect. The established mechanism of action of CsA is to bind to cyclophilin A (PPIA) and restrict dephosphorylation of the NFAT family of genes (not the ARE-responsive genes that are regulated by Nrf2).

Also the reviewer points out a gap in that inhibition of PPIA alone is insufficient to reduce tumor growth, and the authors must amplify this effect with the inhibition of glutaminase. The argument is that PPIA KO results in sensitization of cells to glutamine starvation, but that

inhibition of PPIA alone or glutaminase inhibition alone is insufficient.

Reviewer 2 requested a more thorough glutamine metabolism tracing studies on cell viability to show that PPIA K/O can decrease the flux of glutamine-derived TCA metabolites, which the authors provide. Reviewer 2 also implies that CsA as an immunosuppressive may be an issue and that future in vivo studies should be performed in an immunocompetent mouse model (humanized immune system).

In general, I feel that this paper is interesting scientifically but there is not sufficient discussion of how this is really a biological study with tools but that these tools may not be translatable to the clinic.

Author Response

We thank the Reviewer for the great summary and the highly positive feedback on our study. We have carefully revised our discussions and performed additional experiments to fully address the Reviewer's concerns, as detailed below.

REF 3.2

Reviewer Comment

First— There are still gaps in understanding of mechanism:
How does Nrf2's interaction with PPIA prevent its interaction with Keap1 and subsequent ubiquitination, given that the interaction site for Keap1 and for PPIA are not overlapping? The identified region 169VAQVAPVD176 between the Neh4 and Neh5 domains differs from the Neh2 binding site for Keap1 (76LDEETGEFL84) so the binding does not appear to be competitive with Keap1.

The authors demonstrate through domain truncation experiments to narrow down the potential binding site and then by mutation of P174A they can reduce Nrf2 levels. The cyclophilinA consensus binding motif is FGP X Lp. Would mutation of the Nrf2 sequence between the Neh4 and Neh5 consensus domains to the strongest Cyclophilin binding motif increase Nrf2 levels?

Author Response

We thank the Reviewer for this insightful comment. It is well-known that protein's 3D-structure is critical for understanding the mechanisms of protein-protein interaction. We thus carefully analyzed the NRF2 protein structure predicted by AlphaFold (<https://www.uniprot.org/uniprotkb/Q16236/entry>). Intriguingly, we found that the binding

regions for KEAP1 and PPIA are close to each other in NRF2 protein structure, although these two interaction regions are not overlapped in NRF2 protein sequence. We thus speculated that the binding of PPIA on NRF2 may cause a steric hindrance, which impairs KEAP1 binding and subsequent ubiquitination (Figure R3A).

Moreover, we have replaced the internal sequence of 169VAQVAPVD176 with the consensus PPIA binding motif 169FGPDLPAG176 in NRF2. Intriguingly, we found that these two constructs have comparable protein expression levels and protein stability (Figure R3B and R3C). These findings demonstrate that the region 69VAQVAPVD176 identified in this study is a new, strong binding motif for PPIA.

A

B

C

Figure R3. (A) Human NRF2 protein structure predicted by AlphaFold (AlphaFold ID: AF-Q16236-F1). The region 76LDEETGEFL84 and 169VAQVAPVD176 were colored in cyan. (B) Representative immunoblot analysis of NRF2 protein in 293T cells transfected with NRF2-WT (WT: ¹⁶⁹VAQVAPVD¹⁷⁶) and NRF2-¹⁶⁹MUT (¹⁶⁹MUT: ¹⁶⁹FGPDLPAG¹⁷⁶). (C) Cycloheximide chase assay of NRF2-WT or NRF2-¹⁶⁹MUT construct. Cells were treated with cycloheximide (100 µg/mL) at the indicated time points and then subjected to immunoblot (left panel). Quantitative data from three independent repeats was provided in the right panel.

REF 3.3

Reviewer Comment

Second: Activation of the Nrf2 pathway can occur through adaptive response to electrophiles, such as BG12 (dimethylfumarate). Have the authors tested a combination of dimethyl fumarate with telaglenastat for glutamine metabolism inhibition on the growth of tumor cells in the Xenograft model?

Author Response

We thank the Reviewer for this comment. It is well-known that electrophile dimethylfumarate can oxidize the cysteine residues of KEAP1, subsequently resulting in persistent activation of NRF2 (PMID: 16354693). Previous studies have demonstrated that dimethylfumarate has a cytoprotective role for cancer cells through activation of the NRF2 antioxidant pathway (PMID: 28069874). In a lung cancer mouse model, 61–63% of the tumors in the dimethylfumarate groups were high-grade tumors compared with 52% for the controls ($P < 0.05$), indicating a pro-tumor role of dimethylfumarate in lung cancer (PMID: 25939751). Considering the tumor-supportive effect of dimethylfumarate, we are not tended to examine the anti-tumor activity of the combination of dimethylfumarate with telaglenastat.

References:

- (1) Kobayashi A, Kang MI, Watai Y, et al. Oxidative and electrophilic stresses activate Nrf2 through inhibition of ubiquitination activity of Keap1. *Mol Cell Biol.* 2006;26(1):221-229. doi:10.1128/MCB.26.1.221-229.2006. PMID: 16354693.
- (2) Saidu NE, Noé G, Cerles O, et al. Dimethyl Fumarate Controls the NRF2/DJ-1 Axis in Cancer Cells: Therapeutic Applications. *Mol Cancer Ther.* 2017;16(3):529-539. doi:10.1158/1535-7163.MCT-16-0405. PMID: 28069874.
- (3) To C, Ringelberg CS, Royce DB, et al. Dimethyl fumarate and the oleanane triterpenoids, CDDO-imidazolide and CDDO-methyl ester, both activate the Nrf2 pathway but have opposite effects in the A/J model of lung carcinogenesis. *Carcinogenesis.* 2015;36(7):769-781. doi:10.1093/carcin/bgv061. PMID: 25939751.

REF 3.4

Reviewer Comment

Third: The established mechanism of action of CsA is to bind to cyclophilin A (PPIA) and restrict dephosphorylation of the NFAT family of genes (not the ARE-responsive genes that are regulated by Nrf2). Are there additional cross-talk mechanisms between the NFAT and ARE transcription factor cascades besides the direct binding of cyclophilin A to Nrf2? For example, there are several known links between the ARE-responsive pathway and the NF- κ B responsive pathway. The 'N-factors' in pancreatic cancer: functional relevance of NF- κ B, NFAT and Nrf2 in pancreatic cancer | Oncogenesis (nature.com)

Author Response

We thank the reviewer for raising this intriguing point. An outstanding review by H Kalthoff group have summarized the functional relevance of NF- κ B, NFAT and NRF2 in pancreatic cancer (PMID: 23552468). However, although it is reported that these three pathways interact with each other, the molecule link between NRF2 and NFAT pathway is largely unknown (PMID: 23552468). Of note, our findings raise the possibility that PPIA may act as a molecule node for the cross-talk between NRF2 and NFAT signaling.

To be clear, we have added more explanations in the revised manuscript as following:

"In addition, PPIA is known to regulate transcription factor NFAT, which plays important roles in cancer progression (PMID: 19851316). Our findings raise the possibility that PPIA may act as a molecule node for the cross-talk between NRF2 and NFAT signaling in cancer. Further molecular mechanistic understanding of the reciprocal interactions between NRF2 and NFAT pathway may inspire novel biological insights and therapeutic regimens aimed at targeting tumor progression."

References:

- (1) Arlt A, Schäfer H, Kalthoff H. The 'N-factors' in pancreatic cancer: functional relevance of NF- κ B, NFAT and Nrf2 in pancreatic cancer. *Oncogenesis*. 2012;1(11): e35. Published 2012 Nov 26. doi:10.1038/oncsis.2012.35. PMID: 23552468.
- (2) Mancini M, Toker A. NFAT proteins: emerging roles in cancer progression. *Nat Rev Cancer*. 2009;9(11):810-820. doi:10.1038/nrc2735. PMID: 19851316.

REF 3.5

Reviewer Comment

Fourth: CsA itself is known to cause cancer in high doses (lymphoma and skin cancer), prompting the need for second generation inhibitors. Note FK506 or the more selective L-732531 is a structurally unrelated inhibitor of calcineurin, as is VIVIT (a peptide that interferes with the calcineurin-NFAT interaction). Lastly, there are less toxic analogues of cyclosporin A (ISATX247) The 'N-factors' in pancreatic cancer: functional relevance of NF- κ B, NFAT and Nrf2 in pancreatic cancer - PMC (nih.gov).

Have the authors demonstrated that these other tools work in the same way as CsA on Nrf2 binding, as a potential therapy in the clinic without the cancer-causing attributes of CsA?

It would be interesting to see if the route of administration of these drugs to the lungs (breathing) might have a faster PD response and limit exposures that could cause other cancers.

Author Response

We thank the Reviewer for this comment. The majority of studies regarding the development of malignancy upon CsA treatment stems from organ transplantation research (PMID: 19668519; PMID: 18311436). For instance, it was reported that long-term and high dose of CsA acts as a risk factor for skin cancer development in organ transplant recipients (PMID: 35803453). Whereas, the association of CsA and skin cancer is less clear in non-transplant patients, in which treatment is time-limited and with lower doses (PMID: 29691768; PMID: 35803453). Of note, rheumatoid arthritis patients who are on long-term treatment with CsA have a risk of developing malignancies that is equal to the risk in the general population, and lower than the risk in other rheumatoid arthritis patients (PMID: 10395300). The original CsA developer, Sandoz Ltd. have reported a significantly less skin malignancies with lower doses of CsA (PMID: 21936704). These studies demonstrate that CsA is well-tolerated within the specified dosage ranges. However, more clinical studies are needed to carefully explore the side effects and possible therapeutic benefits of CsA for cancer patients.

The CsA analogue ISATX247 is not commercially available for us. Thus, we turn to use another less toxic CsA analogue NIM811 (PMID: 12065751). Of note, we found that treatment with NIM811 can effectively reduce NRF2 protein levels in A549 and H2030 cells (Figure R4). These results suggest that NIM811 works in the same manner as CsA on NRF2 binding.

We agree with the Reviewer that topical (lung) drug delivery is an intriguing approach to reduce potential sides effects of CsA. However, due to the requirement of specific drug formulation and delivery device, we have not established an inhalation delivery system for CsA in our laboratory.

To address the Reviewer's concern of potential side effects of CsA, we have carefully revised our statements of clinical relevance of our study as following:

“Our study raises the possibility that a deeper understanding of CsA’s mechanism will inspire a more reasonable therapeutic regimen in future clinical investigations using new, optimized PPIA inhibitors to treat patients with NRF2-hyperactivated NSCLC.”

Figure R4. Representative immunoblot analysis of NRF2 protein levels in A549 and H2030 cells treated with NIM811 for 48 h.

References:

- (1) Durnian JM, Stewart RM, Tatham R, Batterbury M, Kaye SB. Cyclosporin-A associated malignancy. *Clin Ophthalmol.* 2007;1(4):421-430. PMID: 19668519
- (2) Väkevä L, Reitamo S, Pukkala E, Sarna S, Ranki A. Long-term follow-up of cancer risk in patients treated with short-term cyclosporine. *Acta Derm Venereol.* 2008;88(2):117-120. doi:10.2340/00015555-0360. PMID: 18311436.
- (3) Rollan MP, Cabrera R, Schwartz RA. Current knowledge of immunosuppression as a risk factor for skin cancer development. *Crit Rev Oncol Hematol.* 2022;177:103754. doi:10.1016/j.critrevonc.2022.103754. PMID: 35803453.
- (4) Howard MD, Su JC, Chong AH. Skin Cancer Following Solid Organ Transplantation: A Review of Risk Factors and Models of Care. *Am J Clin Dermatol.* 2018;19(4):585-597. doi:10.1007/s40257-018-0355-8. PMID: 29691768.
- (5) Landewé RB, van den Borne BE, Breedveld FC, Dijkmans BA. Does cyclosporin A cause cancer? *Nat Med.* 1999;5(7):714. doi:10.1038/10417. PMID: 10395300.
- (6) Muellenhoff MW, Koo JY. Cyclosporine and skin cancer: an international dermatologic perspective over 25 years of experience. A comprehensive review and pursuit to define safe use of cyclosporine in dermatology. *J Dermatolog Treat.* 2012;23(4):290-304. doi:10.3109/09546634.2011.590792. PMID: 21936704.
- (7) Waldmeier PC, Feldtrauer JJ, Qian T, Lemasters JJ. Inhibition of the mitochondrial permeability transition by the nonimmunosuppressive cyclosporin derivative NIM811. *Mol Pharmacol.* 2002;62(1):22-29. doi:10.1124/mol.62.1.22. PMID: 12065751

REF 3.6

Reviewer Comment

Fifth: There are many clinical trials utilizing CB-839 (telaglenastat) along with antibodies for multiple tumor types. There is a disclosure at 2020 ASCO of Phase II trial of KEAPSAKE trial - telalenastat with pembrolizumab (Keytruda)(standard of care chemotherapy) for NSCLC which apparently terminated for lack of clinical benefit. This information could be included in the discussion to clarify the importance of the combination therapy vs CB-839 alone.

A phase II randomized study of telaglenastat, a glutaminase (GLS) inhibitor, versus placebo, in combination with pembrolizumab (Pembro) and chemotherapy as first-line treatment for KEAP1/NRF2-mutated non-squamous metastatic non-small cell lung cancer (mNSCLC). | Journal of Clinical Oncology (ascopubs.org)
Study Record | Beta ClinicalTrials.gov

Author Response

We thank the Reviewer for raising this important point. We have added the overall context of clinical data of CB-839 (telaglenastat) along with other therapeutics in the discussion section in the revised manuscript as following:

“CB-839, a first-in-class glutaminase inhibitor, is actively explored as metabolic intervention for the treatment of multiple types of cancer, but its clinical benefit is limited as monotherapy (PMID: 32907836). Several phase I studies have demonstrated encouraging results for CB-839 as part of a combination regimen (PMID: 340209120). However, a recent phase II study combining CB-839 with pembrolizumab/chemotherapy as first-line treatment for KEAP1/NRF2-mutated metastatic NSCLC have been terminated due to lack of efficacy (NCT04265534). Despite these unsatisfactory outcomes, careful development and rational design of combination therapies with CB-839 might confer clinical benefits.”

References:

- (1) Zhao Y, Feng X, Chen Y, et al. 5-Fluorouracil Enhances the Antitumor Activity of the Glutaminase Inhibitor CB-839 against PIK3CA-Mutant Colorectal Cancers. *Cancer Res.* 2020;80(21):4815-4827. doi:10.1158/0008-5472.CAN-20-0600. PMID: 32907836.
- (2) Yang WH, Qiu Y, Stamatatos O, Janowitz T, Lukey MJ. Enhancing the Efficacy of Glutamine Metabolism Inhibitors in Cancer Therapy. *Trends Cancer.* 2021;7(8):790-804. doi:10.1016/j.trecan.2021.04.003. PMID: 34020912.
- (3) Skoulidis F, et al. A phase II randomized study of telaglenastat, a glutaminase (GLS) inhibitor, versus placebo, in combination with pembrolizumab (Pembro) and chemotherapy as first-line treatment for KEAP1/NRF2-mutated non-squamous metastatic non-small cell lung cancer (mNSCLC). *Journal of Clinical Oncology.* 2020;38(15_suppl): TPS9627-TPS9627. NCT04265534.

REF 3.7

Reviewer Comment

Sixth: Combination therapy with two small molecules would require extensive clinical studies on drug-drug interactions, particularly to avoid tumor cell drug resistance.

Author Response

We thank the Reviewer for this insightful comment. We have added more discussions regarding drug-drug interactions as following:

“Our results may provide a rationale for exploiting CsA to sensitize patients with NRF2-hyperactivated tumors to glutaminase inhibition. However, extensive clinical investigations are required to optimize safe and effective combination therapy regimens of CsA and CB-839 to reduce drug-drug interaction and avoid drug resistance.”

REF 3.8

Reviewer Comment

In general, this paper appears to be scientifically grounded and should be published, but the overall context of the work being performed in the clinic and context of the Phase II studies that have failed would improve the translatability of this work.

Author Response

We thank the Reviewer’s positive comments regarding of our study. To address the Reviewer’s concern, we have added the overall context of clinical investigations in the discussion section to enhance the translatability of our study (**see details in Response 3.6**).

Reviewers' Comments:

Reviewer #2:

Remarks to the Author:

The authors have performed most of the studies in response to the original reviews.

Reviewer #3:

Remarks to the Author:

3.2 The authors argue that by using alpha-fold on Nrf2, which is highly disordered, the binding regions for Keap1 (neh2) and PP1A (Neh4-5 linker) are close in 3-D space despite their distance in primary sequence. While it is an interesting observation, more experiments would need to be done to prove that this model can be trusted. There is no real data for how the two relevant helices near the Neh2 and Neh4-5 linker are placed in 3D space and much of the protein is disordered. But I agree that it is plausible that these regions are close in 3D distance.

Second, when they replace the internal sequence with the consensus PP1A binding motif they find not improved protein expression levels/stability but comparable ones. This, along with the mutagenesis data of P174 that reduced activity, proves that the region between 169-176 is indeed a PP1A binding site.

3.3 It is understandable from the comments made that DMF, as an activator of Nrf2, would not be desirable for tumor inhibition in the clinic, but the proposed experiment was intended to determine if there are direct opposing mechanisms of the Keap1 inhibitors (for preventing Nrf2 degradation by ubiquitination) and PP1A inhibition (for enhancing Nrf2 degradation by some other means). Thank you for pointing out the To et al (2015) paper that suggests that the covalent drug CDDO-methyl-ester reduced the average tumor number and burden compared to control, while DMF had the opposite effect. It seems that there may be poly-pharmacology at play for cancer if these two drugs may work along different signalling pathways. Given the poly-pharmacology of the Keap1 inhibitors, I concede that interpretation of the results of addition of CDDO-methyl-ester to counteract CsA-PP1A inhibition on tumor growth may not be straight-forward.

3.4 I am happy to see this point molecular link of cross talk between the Nrf2 and NFAT signalling pathways being discussed in the revised manuscript.

3.5 I agree with the statement that CsA is well-tolerated within certain dose ranges, but if the goal is to translate to clinic where a dose-range study would be conducted, it would be ideal to identifying alternatives to reduce side effects. I am happy with the revision to the statement of clinical relevance as a result of this work with NIM811.

3.6 I am happy with the revision to the discussion regarding the importance of combination therapies and their historical use in the treatment of metastatic NSCLC. I agree with the sentiment that development of CB-839 in combination with Nrf2 pathway inhibition might confer clinical benefits while prior combination therapies did not.

3.7 I am happy with your inclusion of the importance of drug-drug interaction safety in the discussion.

The authors have thoroughly addressed the majority of my prior concerns and I believe it is ready to be published.